# Controlling Neural Collapse Enhances Out-of-Distribution Detection and Transfer Learning

**Md Yousuf Harun** [1]   **Jhair Gallardo** [1]   **Christopher Kanan** [2]

## Abstract

Out-of-distribution (OOD) detection and OOD generalization are widely studied in Deep Neural Networks (DNNs), yet their relationship remains poorly understood. We empirically show that the degree of Neural Collapse (NC) in a network layer is inversely related with these objectives: stronger NC improves OOD detection but degrades generalization, while weaker NC enhances generalization at the cost of detection. This trade-off suggests that a single feature space cannot simultaneously achieve both tasks. To address this, we develop a theoretical framework linking NC to OOD detection and generalization. We show that entropy regularization mitigates NC to improve generalization, while a fixed Simplex Equiangular Tight Frame (ETF) projector enforces NC for better detection. Based on these insights, we propose a method to control NC at different DNN layers. In experiments, our method excels at both tasks across OOD datasets and DNN architectures.

## 1. Introduction

Out-of-distribution (OOD) detection and OOD generalization are two fundamental challenges in deep learning. OOD detection enables deep neural networks (DNNs) to reject unfamiliar inputs, preventing overconfident mispredictions, while OOD generalization allows DNNs to transfer their knowledge to new distributions. For applications like open-world learning, where a DNN continuously encounters new concepts, both capabilities are essential: OOD detection enables new concepts to be detected, while OOD generalization facilitates forward transfer to improve learning of these new concepts. Despite their importance, these tasks have primarily been studied in isolation. Here, we empirically and theoretically demonstrate a link between both tasks and

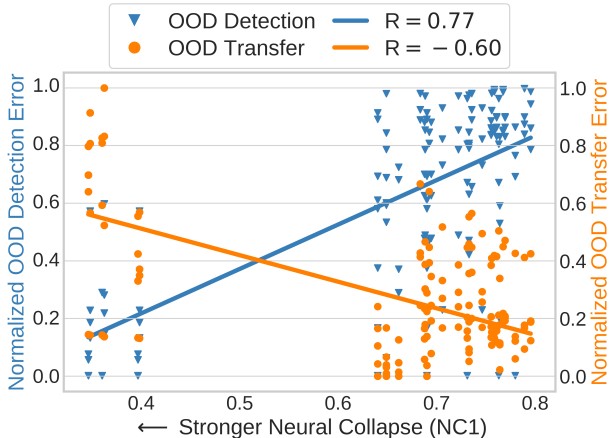

Figure 1: In this paper, we show that there is a close inverse relationship between OOD detection and generalization with respect to the degree of representation collapse in DNN layers. This plot illustrates this relationship for VGG17 pre-trained on ImageNet-100 using four OOD datasets, where we measure collapse and OOD performance for various layers. For OOD detection, there is a strong positive Pearson correlation ($R = 0.77$) with the degree of neural collapse (NC1) in a DNN layer, whereas for OOD generalization, there is a strong negative correlation ($R = -0.60$). We rigorously examine this inverse relationship and propose a method to control NC at different layers.

Neural Collapse (NC), as illustrated in Fig. 1.

NC is a phenomenon where DNNs develop compact and structured class representations (Papyan et al., 2020). While NC was first identified in the final hidden layer, later work has found that it occurs to varying degrees in the last $K$ DNN layers (Rangamani et al., 2023; Harun et al., 2024; Súkeník et al., 2024). NC has a major impact on both OOD detection and generalization. Strong NC improves OOD detection by forming tightly clustered class features that enhance separation between in-distribution (ID) and OOD data (Haas et al., 2023; Wu et al., 2024b; Ming et al., 2022). Conversely, NC impairs OOD generalization by reducing feature diversity, making it harder to transfer knowledge to novel distributions (Kothapalli, 2023; Masarczyk et al., 2023; Harun et al., 2024). However, past work has consid-

[1]Rochester Institute of Technology [2]University of Rochester. Correspondence to: Md Yousuf Harun <mh1023@rit.edu>.

*Proceedings of the 42nd International Conference on Machine Learning*, Vancouver, Canada. PMLR 267, 2025. Copyright 2025 by the author(s).

ered NC in the context of either OOD detection or OOD generalization *individually*, leaving open the question of how NC affects both tasks *simultaneously*. To the best of our knowledge, no prior work has theoretically or empirically examined this relationship.

Here, we establish that the NC exhibited by a DNN layer has an **inverse relationship** with OOD detection and OOD generalization: *stronger NC improves OOD detection but degrades generalization, while weaker NC enhances generalization at the cost of detection performance*. This trade-off suggests that a single feature space cannot effectively optimize both tasks, motivating the need for a novel approach.

We propose a framework that strategically controls NC at different DNN layers to optimize both OOD detection and OOD generalization. We introduce entropy regularization to mitigate NC in the encoder, improving feature diversity and enhancing generalization. Simultaneously, we leverage a fixed Simplex Equiangular Tight Frame (ETF) projector to induce NC in the classification layer, improving feature compactness and enhancing detection. This design enables our DNNs to *decouple representations* for detection and generalization, optimizing both objectives simultaneously.

**Our key contributions are as follows:**

1. We present the first unified study linking *Neural Collapse* to both OOD detection and OOD generalization, empirically demonstrating their inverse relationship and extending analyses of NC beyond the final hidden layer.
2. We develop a theoretical framework that explains how **entropy regularization mitigates NC** to improve OOD generalization. Additionally, we empirically demonstrate that a **fixed Simplex ETF projector enforces NC**, enabling effective OOD detection.
3. In extensive experiments on diverse OOD datasets and DNN architectures, we demonstrate the efficacy of our method compared to baselines. [1]

## 2. Background

### 2.1. OOD Detection

OOD detection methods aim to separate ID and OOD samples by leveraging the differences between their feature representations. Most existing OOD detection methods are *post-hoc*, meaning they apply a scoring function to a model trained exclusively on ID data, without modifying the training process (Salehi et al., 2022). These methods inherently rely on the properties of the learned feature space to distinguish ID from OOD samples.

Post-hoc detection techniques can be broadly categorized based on the source of their confidence estimates. Density-based methods model the ID distribution probabilistically

and classify low-density test points as OOD (Lee et al., 2018; Zisselman & Tamar, 2020; Choi et al., 2018; Jiang et al., 2023). More commonly, confidence-based approaches estimate OOD likelihood using model outputs (Hendrycks & Gimpel, 2016; Liang et al., 2017; Liu et al., 2020), feature statistics (Sun et al., 2021; Zhu et al., 2022a; Sun et al., 2022), or gradient-based information (Huang et al., 2021; Wu et al., 2024a; Lee et al., 2023; Igoe et al., 2022).

Since post-hoc methods depend on the representations learned during ID training, their effectiveness is fundamentally constrained by the quality of those features (Roady et al., 2020). Highly compact, well-separated ID representations generally improve OOD detection by reducing feature overlap with OOD samples. For example, Haas et al. (2023) demonstrated that $L_2$ normalization of penultimate-layer features induces NC, enhancing ID-OOD separability. Similarly, Wu et al. (2024b) introduced a regularization loss that enforces orthogonality between ID and OOD representations, leveraging NC-like properties to improve detection. NECO (Ammar et al., 2024), a post-hoc OOD detection method, leverages NC and the orthogonality between ID and OOD samples to achieve state-of-the-art performance. However, unlike our approach, NECO and other methods do not focus on OOD generalization or representation learning.

Another representation learning approach is to learn representations explicitly tailored for OOD detection by incorporating OOD samples during training (Wu et al., 2024b; Bai et al., 2023; Katz-Samuels et al., 2022; Ming et al., 2022). These methods encourage models to assign lower confidence (Hendrycks et al., 2018) or higher energy (Liu et al., 2020) to OOD inputs. However, this approach presents significant challenges, as the space of possible OOD data is essentially infinite, making it impractical to represent all potential OOD variations. Moreover, strong OOD detection performance often comes at the cost of degraded OOD generalization (Zhang et al., 2024), as representations optimized for separability may lack the diversity needed for adaptation to novel distributions.

### 2.2. Transfer Learning and OOD Generalization

Transfer learning and OOD generalization methods focus on learning features that remain effective across distribution shifts. Robust transfer is particularly important in open-world learning scenarios, where models must not only adapt to new distributions but also improve sample efficiency over time, a key requirement for continual learning. To facilitate generalization, techniques such as feature alignment (Li et al., 2018b; Ahuja et al., 2021; Zhao et al., 2020; Ming et al., 2024), ensemble/meta-learning (Balaji et al., 2018; Li et al., 2018a; 2019; Bui et al., 2021), robust optimization (Rame et al., 2022; Cha et al., 2021; Krueger et al., 2021; Shi et al., 2021), data augmentation (Nam et al., 2021; Nuriel et al., 2021; Zhou et al., 2020), and feature disentan-

---

glement (Zhang et al., 2022) have been proposed.

Key properties of learned features significantly impact generalization to unseen distributions. Studies examining factors that affect OOD generalization emphasize that feature diversity is essential for robustness (Masarczyk et al., 2023; Harun et al., 2024; Kornblith et al., 2021; Fang et al., 2024; Ramanujan et al., 2024; Kolesnikov et al., 2020; Vishniakov et al., 2024). Notably, recent work (Kothapalli, 2023; Masarczyk et al., 2023; Harun et al., 2024) suggests that progressive feature compression in deeper layers, linked to NC emergence, can hinder OOD generalization by reducing representation expressivity.

### 2.3. Neural Collapse

As noted earlier, NC arises when class features become tightly clustered, often converging toward a Simplex ETF (Papyan et al., 2020; Kothapalli, 2023; Zhu et al., 2021; Han et al., 2022). Initially, NC was studied primarily in the final hidden layer, but later work demonstrated that NC manifests to varying degrees in earlier layers as well (Rangamani et al., 2023; Harun et al., 2024). In image classification experiments, Harun et al. (2024) showed that the degree of intermediate NC is heavily influenced by the properties of the training data, including the number of ID classes, image resolution, and the use of augmentations.

NC can be characterized by four main properties:

1. **Feature Collapse** ($\mathcal{NC}1$): Features within each class concentrate around a single mean, exhibiting minimal intra-class variability.
2. **Simplex ETF Structure** ($\mathcal{NC}2$): When centered at the global mean, class means lie on a hypersphere with maximal pairwise distances, forming a Simplex ETF.
3. **Self-Duality** ($\mathcal{NC}3$): The last-layer classifiers align tightly with their corresponding class means, creating a nearly self-dual configuration.
4. **Nearest Class Mean Decision** ($\mathcal{NC}4$): Classification behaves like a nearest-centroid scheme, assigning classes based on proximity to class means.

While NC's structured representations can aid OOD detection by ensuring strong class separability (Haas et al., 2023; Wu et al., 2024b), the same compression may limit the feature diversity needed for generalization. One proposed explanation is the *Tunnel Effect Hypothesis* (Masarczyk et al., 2023), which suggests that as features become increasingly compressed in deeper layers, generalization to unseen distributions is impeded.

## 3. Problem Definition

We consider a supervised multi-class classification problem where the input space is $\mathcal{X}$ and the label space is $\mathcal{Y} = \{1, 2, \ldots, K\}$. A model parameterized by $\theta$, $f_\theta : \mathcal{X} \to \mathbb{R}^K$, is trained on ID data drawn from $P^{\text{ID}}_{\mathcal{X}\mathcal{Y}}$, to produce logits, $f_\theta(x)$ which are used to predict labels. For robust operation in real-world scenarios, the model must classify samples from $P^{\text{ID}}_{\mathcal{X}\mathcal{Y}}$ correctly and identify OOD samples from $P^{\text{OOD}}_{\mathcal{X}\mathcal{Y}'}$ which represents the distribution with no overlap with the ID label space, i.e., $\mathcal{Y} \cap \mathcal{Y}' = \emptyset$.

At test time, the objective of OOD detection is to determine whether a given sample $x$ originates from the ID or an OOD source. This can be achieved by using a threshold-based decision rule through level set estimation, defined as:

$$G_\lambda(x) = \begin{cases} \text{ID}, & S(x) \geq \lambda \\ \text{OOD}, & S(x) < \lambda \end{cases}$$

where $S(\cdot)$ is a scoring function, and samples with $S(x) \geq \lambda$ are classified as ID, while those with $S(x) < \lambda$ are classified as OOD. $\lambda$ denotes the threshold.

On the OOD generalization part, the objective is to build a model $f_\theta^* : \mathcal{X} \to \mathbb{R}^K$, using the ID data such that it learns *transferable* representations and becomes adept at both ID task and OOD downstream tasks. This is a challenging problem since we do not have access to OOD data during training. In both OOD detection and OOD generalization, the label space is disjoint between ID and OOD sets.

**Differences from prior works.** Prior works (Zhang et al., 2024; Wang & Li, 2024; Bai et al., 2023) focusing on OOD detection and OOD generalization, define the problem differently than us. For OOD detection, they use "*semantic OOD*" data from $P^{\text{SEM}}_{\mathcal{X}\mathcal{Y}'}$ that has no semantic overlap with known label space from $P^{\text{ID}}_{\mathcal{X}\mathcal{Y}}$, i.e., $\mathcal{Y} \cap \mathcal{Y}' = \emptyset$. However, for OOD generalization, they use "*covariate-shifted OOD*" data from $P^{\text{COV}}_{\mathcal{X}\mathcal{Y}}$, that has the same label space as $P^{\text{ID}}_{\mathcal{X}\mathcal{Y}}$ but with shifted marginal distributions $P^{\text{COV}}_{\mathcal{X}}$ due to noise or corruption. Furthermore, they use additional semantic OOD training data, $P^{\text{SEM}}_{\mathcal{X}\mathcal{Y}'}$ during the training phase. Our problem definition is fundamentally more challenging and practical than the prior works because: (1) we aim to detect semantic OOD samples, $P^{\text{SEM}}_{\mathcal{X}\mathcal{Y}'}$ without access to auxiliary OOD data during training, and (2) we aim to generalize to semantic OOD samples that belong to novel semantic categories.

**Evaluation Metrics.** We define ID generalization error ($\mathcal{E}_{\text{ID}}$), OOD generalization error ($\mathcal{E}_{\text{GEN}}$), and OOD detection error ($\mathcal{E}_{\text{DET}}$) as follows:

1. $\downarrow \mathcal{E}_{\text{ID}} := 1 - \mathbb{E}_{(\bar{x},y) \sim P^{\text{ID}}} \left( \mathbb{I}\{\hat{y}(f_\theta(\bar{x})) = y\} \right)$,
2. $\downarrow \mathcal{E}_{\text{GEN}} := 1 - \mathbb{E}_{(\bar{x},y) \sim P^{\text{OOD}}} \left( \mathbb{I}\{\hat{y}(f_\theta(\bar{x})) = y\} \right)$,
3. $\downarrow \mathcal{E}_{\text{DET}} := \mathbb{E}_{\bar{x} \sim P^{\text{OOD}}} \left( \mathbb{I}\{G_\lambda(\bar{x}) = \text{ID}\} \right)$,

where $\mathbb{I}\{\cdot\}$ denotes the indicator function, and the arrows indicate that lower is better. For OOD detection, ID samples are considered positive. FPR95 (false positive rate at 95% true positive rate) is used as $\mathcal{E}_{\text{DET}}$. Details are in Appendix.

**OOD Detection.** Following earlier work (Sun et al., 2021;

Liu et al., 2020), we consider the energy-based scoring (Liu et al., 2020) since it operates with logits and does not require any fine-tuning or hyper-parameters. Scoring in energy-based models is defined as

$$S(x) = -\log \sum_{k=1}^{K} \exp\left(f_k\left(x\right)\right)$$

where the $k$-th logit, $f_k(x)$, denotes the model's confidence for assigning $x$ to class $k$. Note that Liu et al. (2020) uses the negative energy, meaning that OOD samples should obtain high energy, hence low $S(x)$. See Fig. 6 as an example.

**OOD Generalization.** For evaluating OOD generalization, we consider linear probing which is widely used to evaluate the transferability of learned embeddings to OOD datasets (Alain & Bengio, 2016; Masarczyk et al., 2023; Zhu et al., 2022b; Waldis et al., 2024; Grill et al., 2020; He et al., 2020). For a given OOD dataset, we extract embeddings from a pre-trained model. A linear probe (MLP classifier) is then attached to map these embeddings to OOD classes. The probe is trained and evaluated on OOD data.

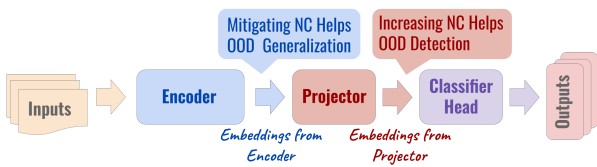

Figure 2: **Implication of Neural Collapse.** Mitigating NC in the encoder enhances OOD generalization whereas increasing NC in the projector improves OOD detection.

# 4. Controlling Neural Collapse

Typically, penultimate-layer embeddings from a pre-trained DNN are used for downstream tasks. However, using the same embedding space for both OOD detection and OOD generalization is suboptimal due to their conflicting objectives. We therefore propose separate embedding spaces at different layers—one for OOD detection, another for OOD generalization. Specifically, we attach a projector network $g(\cdot)$ to the DNN backbone $f(\cdot)$ (the encoder) and add a classifier head $h(\cdot)$ on top. Given an input $\mathbf{x}$, the encoder outputs $\mathbf{f} = f(\mathbf{x})$, e.g., a 512-dimensional vector for ResNet18. The projector then maps $\mathbf{f}$ to $\mathbf{g} = g(\mathbf{f})$, and finally the classifier produces logits $\mathbf{h} = h(\mathbf{g}) \in \mathbb{R}^K$.

The encoder is trained to prevent NC and encourage transferable representations for OOD generalization, while the projector is designed to induce NC, producing collapsed representations beneficial for OOD detection. A high-level illustration is provided in Fig. 2. For OOD detection and ID classification tasks, the entire network ($h \circ g \circ f$) is utilized, assuming projector embedding $\mathbf{g}$ is most discriminative

among all layers. Whereas the encoder alone is utilized for OOD generalization, assuming encoder embedding $\mathbf{f}$ is most transferable among all layers. In the following subsections, we will portray how we can build these collapsed and transferable representations.

### 4.1. Entropy Regularization Mitigates Neural Collapse

In this section, we provide a theoretical justification for using an entropy regularizer to prevent or mitigate *intermediate neural collapse* (NC1) in deep networks. By "intermediate" we mean that the collapse occurs in hidden layers.

**Setup and Notation.** Let $L$ be the total number of layers in our network, and $\ell \in \{1, 2, \ldots, L\}$ the intermediate layer index. We denote the embedding (activation) in layer $\ell$ for the $i$-th sample $\mathbf{x}_i$ as $\mathbf{z}_{\ell,i} = f_\ell(\mathbf{x}_i)$, where $\mathbf{z}_{\ell,i} \in \mathbb{R}^{d_\ell}$. Suppose we have $K$ classes, labeled by $1, \ldots, K$. We can view the random variable $\mathbf{Z}_\ell$ (the layer-$\ell$ embeddings) as distributed under the data distribution according to

$$p_\ell(\mathbf{z}) = \sum_{k=1}^{K} \pi_k \, p_{\ell,k}(\mathbf{z})$$

where $\pi_k = \Pr(y = k)$ is the class prior, and $p_{\ell,k}(\mathbf{z})$ is the class–conditional distribution of $\mathbf{Z}_\ell$ for label $k$.

**Intermediate Neural Collapse (NC1).** Empirically, *neural collapse* is observed when the within-class covariance of these embeddings shrinks as training proceeds. Formally, for each class $k$, the distribution $p_{\ell,k}$ concentrates around its class mean $\boldsymbol{\mu}_{\ell,k} \in \mathbb{R}^{d_\ell}$, resulting in:

$$\mathrm{Trace}(\boldsymbol{\Sigma}_{\ell,k}) \to 0, \quad \text{where} \quad \boldsymbol{\Sigma}_{\ell,k} = \mathrm{Cov}\big(\mathbf{Z}_\ell \mid y = k\big).$$

Although often highlighted in the *penultimate* layer, such collapse can appear across the final layers of a DNN (Rangamani et al., 2023; Harun et al., 2024).

**Differential Entropy and Collapsing Distributions.** For a continuous random variable $\mathbf{Z}_\ell \in \mathbb{R}^{d_\ell}$ with density $p_\ell(\mathbf{z})$, the *differential entropy* is given by

$$H\big(p_\ell\big) = -\int_{\mathbb{R}^{d_\ell}} p_\ell(\mathbf{z}) \log p_\ell(\mathbf{z}) \, d\mathbf{z}$$

It is well known that if $p_{\ell,k}$ collapses to a delta (or near-delta) around $\boldsymbol{\mu}_{\ell,k}$, then $H(p_{\ell,k}) \to -\infty$ (Cover, 1999). Consequently, a *mixture* of such collapsing class–conditional distributions also attains arbitrarily negative entropy. The following proposition formalizes this point.

**Proposition 4.1** (Entropy under Class–Conditional Collapse)**.** *Consider a mixture distribution* $p_\ell(\mathbf{z}) = \sum_{k=1}^{K} \pi_k \, p_{\ell,k}(\mathbf{z})$ *on* $\mathbb{R}^{d_\ell}$. *Suppose that, for each $k$, $p_{\ell,k}$ becomes arbitrarily concentrated around a single point* $\boldsymbol{\mu}_{\ell,k}$.

*In the limit where each $p_{\ell,k}$ approaches a Dirac delta, the differential entropy $H(p_\ell)$ diverges to $-\infty$.*

*Proof Sketch.* If each $p_{\ell,k}$ is a family of densities approaching $\delta(\mathbf{z} - \boldsymbol{\mu}_{\ell,k})$, the individual entropies $H(p_{\ell,k})$ go to $-\infty$. The entropy of the mixture can be bounded above by the weighted sum of $H(p_{\ell,k})$ plus a constant that depends on the mixture overlap. Hence, the overall mixture entropy also diverges to $-\infty$. □

Appendix E contains the detailed proof.

**Entropy Regularization to Mitigate Collapse.** We see from Proposition 4.1 that if *all* class–conditional distributions collapse to near-deltas, the layer's overall density $p_\ell(\mathbf{z})$ has differential entropy $H(p_\ell) \to -\infty$. Since standard classification objectives can favor very tight class clusters (e.g., to sharpen decision boundaries), one can counteract this by *maximizing $H(p_\ell)$*.

Concretely, we augment the training loss $\mathcal{L}_{\text{cls}}(\theta)$ with a negative–entropy penalty:

$$\mathcal{L}_{\text{total}}(\theta) = \mathcal{L}_{\text{cls}}(\theta) - \alpha H\big(p_\ell(\mathbf{z} \mid \theta)\big), \qquad (1)$$

where $\alpha > 0$ is a hyperparameter. As $\boldsymbol{\Sigma}_{\ell,k} \to 0$ would force $H(p_\ell)$ to $-\infty$ (cf. Proposition 4.1), the additional term $-\alpha H(p_\ell)$ becomes unboundedly large. Therefore, the model is compelled to maintain *nonzero within–class variance* for each class distribution, preventing complete layer collapse.

Since we do not have direct access to $p_\ell(\mathbf{z})$, we need to estimate $H(p_\ell)$ using a data-driven density estimation approach. In particular, prior work (Kozachenko & Leonenko, 1987; Beirlant et al., 1997) shows that the differential entropy can be estimated by nearest neighbor distances.

Given a batch of $N$ random representations $\{\mathbf{z}_n\}_{n=1}^N$, the nearest neighbor entropy estimate is given by

$$H\big(p_\ell\big) \approx \frac{1}{N} \sum_{n=1}^N \log\left(N \min_{i \in [N], i \neq n} \|\mathbf{z}_n - \mathbf{z}_i\|_2\right) + \ln 2 + \text{EC}$$

where EC denotes the Euler constant. For our purposes, we can simplify the entropy maximization objective by removing affine terms, resulting in the following loss function:

$$\mathcal{L}_{\text{reg}}(\theta) = -\frac{1}{N} \sum_{n=1}^N \log\left(\min_{i \in [N], i \neq n} \|\bar{\mathbf{z}}_n - \bar{\mathbf{z}}_i\|_2\right)$$

Total loss becomes: $\mathcal{L}_{\text{total}}(\theta) = \mathcal{L}_{\text{cls}}(\theta) + \alpha\, \mathcal{L}_{\text{reg}}(\theta)$. Intuitively, $\mathcal{L}_{\text{reg}}$ maximizes the distance between the nearest pairs in the batch, encouraging an even spread of representations across the embedding space. The pairwise distances

can be sensitive to outliers with large magnitudes. Therefore, in our method, the loss operates on the hyperspherical embedding space with the unit norm, i.e., $\bar{\mathbf{z}} = \mathbf{z}/\|\mathbf{z}\|_2$. Note that unlike $\mathcal{L}_{\text{cls}}$ acting on classifier head, $\mathcal{L}_{\text{reg}}$ is applied in encoder for mitigating NC.

Although various loss functions including cross-entropy (CE) and mean-squared-error (MSE) lead to NC, others produce less transferable features than CE (Zhou et al., 2022; Kornblith et al., 2021). We also find that CE outperforms MSE in both OOD detection and OOD generalization (see Table 9). Therefore, we consider CE loss for $\mathcal{L}_{\text{cls}}$ in Equation 1. It has been found that using label smoothing with CE loss intensifies NC properties when compared with the regular CE loss (Zhou et al., 2022; Kornblith et al., 2021). Therefore, we use label smoothing with CE loss to expedite NC properties in the projector and classifier head.

In addition to $\mathcal{L}_{\text{reg}}$ mitigating NC, we consider alternatives to batch normalization (BN). In the context of learning transferable representations in the encoder, batch dependency, especially using BN, is sub-optimal as OOD data statistically differs from ID data. Therefore, for all layers in the encoder, we replace batch normalization with a batch-independent alternative, particularly, a combination of group normalization (GN) (Wu & He, 2018) and weight standardization (WS) (Qiao et al., 2019) to enhance OOD generalization.

### 4.2. Simplex ETF Projector for Inducing Collapse

When a DNN enters into NC phase, the class-means converge to a simplex ETF (equinorm and maximal equiangularity) in collapsed layers (NC2 criterion). This implies that fixing the collapsed layers to be ETFs does not impair ID performance (Rangamani et al., 2023; Zhu et al., 2021). In this work, we induce NC in the projector to improve OOD detection performance. We do it by fixing the projector to be simplex ETF, acting as an architectural inductive bias.

Our projector comprises two MLP layers sandwiched between encoder and classifier head. We set the projector weights to simplex ETFs and keep them frozen during training. In particular, each MLP layer is set to be a rank $\mathcal{D} - 1$ simplex ETF, where $\mathcal{D}$ denotes width or output feature dimension. The rank $\mathcal{D}$ canonical simplex ETF is:

$$\sqrt{\frac{\mathcal{D}}{\mathcal{D} - 1}}\left(\mathbf{I}_{\mathcal{D}} - \frac{1}{\mathcal{D}}\mathbf{1}_{\mathcal{D}}\mathbf{1}_{\mathcal{D}}^T\right)$$

Details on the projector are given in Appendix A.1. We further apply $L_2$ normalization to the output of the projector since it constraints features to achieve equinormality and helps induce early neural collapse (Haas et al., 2023).

While prior work has found that incorporating a projector improves transfer in supervised learning (Wang et al., 2022a), the objective of our projector is to impede transfer.

The difference is that a projector is typically trained along with the backbone, whereas in our method the projector is configured as Simplex ETF and kept frozen during training.

## 5. Experimental Setup

**Datasets.** For ID dataset, we use ImageNet-100 (Tian et al., 2020)- a subset (100 classes) of ImageNet-1K (Russakovsky et al., 2015). To assess OOD generalization and OOD detection, we study eight commonly used OOD datasets: NINCO (Bitterwolf et al., 2023), ImageNet-R (Hendrycks et al., 2021), CIFAR-100 (Krizhevsky & Hinton, 2014), Oxford 102 Flowers (Nilsback & Zisserman, 2008), CUB-200 (Wah et al., 2011), Aircrafts (Maji et al., 2013), Oxford-IIIT Pets (Parkhi et al., 2012), and STL-10 (Coates et al., 2011). Dataset details are given in Appendix B.

**DNN Architectures.** We train and evaluate a representative set of DNN architectures including VGG (Simonyan & Zisserman, 2015), ResNet (He et al., 2016), and ViT (Dosovitskiy et al., 2020). In total, we experiment with five backbones: VGG17, ResNet18, ResNet34, ViT-Tiny, and ViT-Small. Our projector is composed of two MLP layers for all DNN architectures. Details are given in Appendix A.1.

**NC Metrics ($\mathcal{NC}1$–$\mathcal{NC}4$).** We use four metrics, $\mathcal{NC}1$, $\mathcal{NC}2$, $\mathcal{NC}3$, and $\mathcal{NC}4$, as described in (Zhu et al., 2021; Zhou et al., 2022), to evaluate the NC properties of the DNN features and classifier. These metrics correspond to four NC properties outlined in Sec. 2.3. Note that $\mathcal{NC}1$ is the most dominant indicator of neural collapse. We describe each NC metric in detail in the Appendix C.

**Training Details.** In our main experiments, we train different DNN architectures e.g., VGG17, ResNet18, and ViT-T on ImageNet-100 for 100 epochs. The Entropy regularization loss $\mathcal{L}_{\text{reg}}$ is modulated with $\alpha = 0.05$. We use AdamW (Loshchilov, 2017) optimizer and cosine learning rate scheduler with a linear warmup of 5 epochs. For a batch size of 512, we set the learning rate to $6 \times 10^{-3}$ for VGG17, $0.01$ for ResNet18, and $8 \times 10^{-4}$ for ViT-T. For all models, we set the weight decay to $0.05$ and the label smoothing to $0.1$. In all our experiments, we use $224 \times 224$ images. And, we use random resized crop and random horizontal flip augmentations. Linear probes are attached to the encoder and projector layers of a pre-trained model and trained on extracted embeddings of OOD data using the AdamW optimizer and CE loss for 30 epochs. Additional implementation details are given in Appendix A.

**Baselines.** Recent work defines the problem differently where they focus on OOD detection and covariate OOD generalization (same labels but different input distribution). Our problem setup focuses on OOD detection and semantic OOD generalization (different labels and different input distribution). Adapting other methods to our problem

Table 1: **Main Results (Encover Vs. Projector).** Various DNNs are trained on ImageNet-100 dataset (ID) and evaluated on eight OOD datasets. All models incorporate entropy regularization and the ETF projector to control NC. Reported $\mathcal{E}_{\text{GEN}}$ (%) and $\mathcal{E}_{\text{DET}}$ (%) are averaged over eight OOD datasets. **A lower $\mathcal{NC}$ indicates stronger neural collapse.** $+\Delta_{E \to P}$ and $-\Delta_{E \to P}$ indicate % increase and % decrease respectively, when changing from the encoder ($E$) to projector ($P$).

| Model | $\mathcal{E}_{\text{ID}}$ ↓ | Neural Collapse | | | | $\mathcal{E}_{\text{GEN}}$ Avg. ↓ | $\mathcal{E}_{\text{DET}}$ Avg. ↓ |
| --- | --- | --- | --- | --- | --- | --- | --- |
| | | $\mathcal{NC}1$ | $\mathcal{NC}2$ | $\mathcal{NC}3$ | $\mathcal{NC}4$ | | |
| **VGG17** | | | | | | | |
| Projector | **12.62** | 0.393 | 0.490 | 0.468 | 0.316 | 66.36 | **65.10** |
| Encoder | 15.52 | 2.175 | 0.603 | 0.616 | 5.364 | **41.85** | 87.62 |
| $\Delta_{E \to P}$ | -18.69 | -81.93 | -18.74 | -24.03 | -94.11 | +58.57 | -25.70 |
| **ResNet18** | | | | | | | |
| Projector | **16.14** | 0.341 | 0.456 | 0.306 | 0.540 | 63.08 | **69.70** |
| Encoder | 20.14 | 1.762 | 0.552 | 0.555 | 10.695 | **47.72** | 86.17 |
| $\Delta_{E \to P}$ | -19.86 | -80.65 | -17.39 | -44.86 | -94.95 | +32.19 | -19.11 |
| **ViT-T** | | | | | | | |
| Projector | **32.04** | 2.748 | 0.609 | 0.798 | 1.144 | 63.53 | **83.16** |
| Encoder | 33.94 | 5.769 | 0.748 | 0.847 | 2.332 | **52.63** | 90.89 |
| $\Delta_{E \to P}$ | -5.60 | -52.37 | -18.58 | -5.79 | -50.94 | +20.71 | -8.50 |

setup will require major modifications, hence we cannot compare directly with them. We compare the proposed method with baselines that do not use any of our mechanisms e.g., entropy regularization or fixed simplex ETF projector. Additionally, in Sec. 6.5, we include a comparison with NECO (Ammar et al., 2024), a state-of-the-art OOD detection method that leverages NC properties.

## 6. Experimental Results

Sec. 6.1 shows how controlling NC improves representations for OOD detection and generalization. We compare with baselines in Sec. 6.2, and analyze the roles of entropy regularization and the ETF projector in Sec. 6.3 and Sec. 6.4, respectively. Sec. 6.5 presents a comparison with NECO, and Sec. 6.6 summarizes additional results.

### 6.1. Impact of Controlling NC

We investigate whether controlling NC improves OOD detection and generalization by examining NC properties in the encoder and projector. Table 1 summarizes the results across eight OOD datasets. The projector, which exhibits lower NC values (i.e., stronger NC), achieves superior OOD detection (7.73%–22.52% margin) and lower ID error compared to the encoder. In contrast, the encoder's higher NC values (i.e., weaker NC) lead to better OOD generalization (10.90%–24.51% margin) than the projector. Comprehensive results across OOD datasets are given in Appendix F.

We also visualize the encoder and projector embeddings in Fig 3 and 8 for deeper insights. Unlike encoder embeddings, projector embeddings cluster tightly around class means

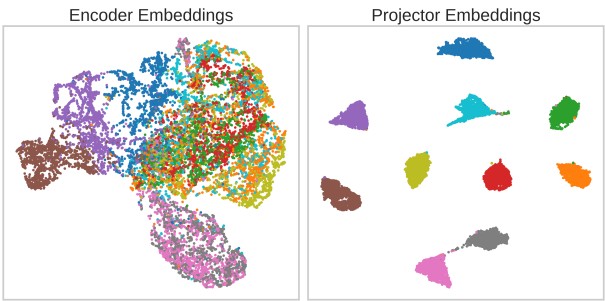

Figure 3: **UMAP Visualization of Embedding.** The projector embeddings exhibit much greater NC ($\mathcal{NC}1 = 0.393$) than the encoder embeddings ($\mathcal{NC}1 = 2.175$) as indicated by the formation of compact clusters around class means. For clarity, we highlight 10 ImageNet classes by distinct colors. For this, we use ImageNet-100 pre-trained VGG17.

Table 2: **Comparison with Baseline.** Various DNNs e.g., VGG17, ResNet18, and ViT-T are trained on ImageNet-100 dataset (ID). Baseline models do not incorporate mechanisms like entropy regularization or the ETF projector to control NC. NC metrics are computed using the penultimate-layer embeddings. Reported $\mathcal{E}_{\text{GEN}}$ (%) and $\mathcal{E}_{\text{DET}}$ (%) are averaged over eight OOD datasets.

| Model | $\mathcal{E}_{\text{ID}}$ $\downarrow$ | Neural Collapse | | | | $\mathcal{E}_{\text{GEN}}$ Avg. $\downarrow$ | $\mathcal{E}_{\text{DET}}$ Avg. $\downarrow$ |
| | | $\mathcal{NC}1$ | $\mathcal{NC}2$ | $\mathcal{NC}3$ | $\mathcal{NC}4$ | | |
|---|---|---|---|---|---|---|---|
| VGG17 | 12.18 | 0.766 | 0.705 | 0.486 | 37.491 | 49.54 | 94.92 |
| **+Ours** | 12.62 | 0.393 | 0.490 | 0.468 | 0.316 | **41.85** | **65.10** |
| ResNet18 | 15.38 | 1.11 | 0.658 | 0.590 | 31.446 | 49.42 | 97.40 |
| **+Ours** | 16.14 | 0.341 | 0.456 | 0.306 | 0.540 | **47.72** | **69.70** |
| ViT-T | 31.78 | 2.467 | 0.657 | 0.601 | 1.015 | 52.68 | 90.17 |
| **+Ours** | 32.04 | 2.748 | 0.609 | 0.798 | 1.144 | 52.63 | **83.16** |

(reflecting stronger NC1). Additionally, Fig. 5a in the Appendix shows that the projector exhibits greater ID–OOD separation than the encoder. Finally, Fig. 5b and 6 in the Appendix show that the energy score distribution reveals the projector separates ID from OOD more effectively than the encoder across multiple datasets. These observations explain why the projector excels at OOD detection by exploiting more collapsed features.

Finally, we analyze different layers of VGG17 and ResNet18 models and find that increasing NC strongly correlates with lower OOD detection error and reducing NC strongly correlates with lower OOD generalization error, as shown in Fig. 1 and 7. Our experimental results validate that *controlling NC effectively enhances OOD detection and OOD generalization abilities.*

### 6.2. Comparison with Baseline

We want to check how standard DNNs perform without any mechanisms to control NC. As depicted in Table 2, different

Table 3: **Impact of Entropy Regularization.** VGG17 models are trained on ImageNet-100 dataset (ID) and evaluated on eight OOD datasets. Both models share the same architecture and use an ETF projector; the difference lies solely in the use of entropy regularization. NC metrics are measured with encoder embeddings where entropy regularization is applied or omitted. Reported $\mathcal{E}_{\text{GEN}}$ (%) and $\mathcal{E}_{\text{DET}}$ (%) correspond to the encoder and projector, respectively and are averaged across eight OOD datasets.

| Method | $\mathcal{E}_{\text{ID}}$ $\downarrow$ | Neural Collapse $\uparrow$ | | | | $\mathcal{E}_{\text{GEN}}$ Avg. $\downarrow$ | $\mathcal{E}_{\text{DET}}$ Avg. $\downarrow$ |
| | | $\mathcal{NC}1$ | $\mathcal{NC}2$ | $\mathcal{NC}3$ | $\mathcal{NC}4$ | | |
|---|---|---|---|---|---|---|---|
| No Reg. | 13.46 | 1.31 | 0.72 | 0.62 | 5.18 | 44.56 | 67.46 |
| **Reg.** | **12.62** | **2.18** | 0.61 | 0.62 | 5.36 | **41.85** | **65.10** |

DNNs including VGG17, ResNet18, and ViT-T land on higher OOD detection error and OOD generalization error indicating that representations learned by these models cannot achieve both OOD detection and OOD generalization abilities. In contrast, our method shows significant improvements over these baselines. While being competitive in ID performance, our method controls NC unlike the baselines, and achieves better performance in OOD tasks. Particularly, OOD generalization is improved by $1.70\% - 7.69\%$ (absolute) and OOD detection is improved by $7.01\% - 29.82\%$ (absolute). More comprehensive results across eight OOD datasets are given in Table 18 in the Appendix.

### 6.3. Entropy Regularization Mitigates NC

At first, we measure entropy and NC1 across all VGG17 layers and observe that there lies a strong correlation between entropy and NC1 (Pearson correlation 0.88). As illustrated in Fig. 4, stronger NC correlates with lower entropy whereas weaker NC correlates with higher entropy. This empirically demonstrates why using entropy regularization mitigates NC in the encoder. Next, we compare two identical VGG17 models, one uses entropy regularization and another omits it. The results are summarized in Table 3. Entropy regularization mitigates NC in the encoder, as evidenced by its higher $\mathcal{NC}1$, and achieves better performance in all criteria compared to the model without entropy regularization.

An implication of NC is that the collapsed layers exhibit lower rank in the weights and representations (Rangamani et al., 2023). In additional analyses provided in the Appendix G, we observe that our entropy regularization implicitly encourages higher rank in the encoder embeddings and helps reduce dependence between dimensions (thereby promoting mutual independence).

### 6.4. Fixed Simplex ETF Projector Induces NC

We train two identical models the same way (same hyperparameters and training protocol), one of them uses a regular

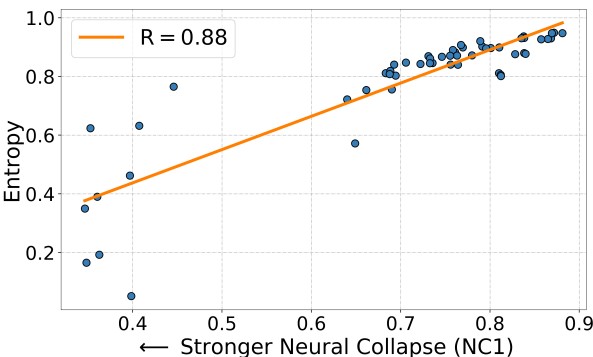

Figure 4: Neural collapse (NC1) correlates with entropy. The stronger the neural collapse, the lower the entropy and vice-versa. This suggests that increasing the entropy of the encoder's embeddings can help mitigate NC and enhance OOD generalization. Similar to Fig. 1, we analyze different layers of VGG17 networks that are pre-trained on the ImageNet-100 (ID) dataset. $R$ denotes the Pearson correlation coefficient.

Table 4: **Impact of Fixed ETF Projector and $L_2$ Normalization on NC.** The evaluation is based on **projector embeddings** of ImageNet-100 pre-trained VGG17 networks. A lower $\mathcal{NC}$ indicates higher neural collapse. Our model (highlighted) uses a fixed ETF projector with $L_2$ normalization, whereas the baseline **Plastic** uses a trainable projector with $L_2$ normalization, and the baseline **No L$_2$ norm** uses a fixed ETF projector but omits $L_2$ normalization. Reported $\mathcal{E}_{\text{DET}}$ (%) is averaged over eight OOD datasets.

| Projector | $\mathcal{E}_{\text{ID}}$ ↓ | Neural Collapse ↓ | | | | $\mathcal{E}_{\text{DET}}$ |
|---|---|---|---|---|---|---|
| | | $\mathcal{NC}1$ | $\mathcal{NC}2$ | $\mathcal{NC}3$ | $\mathcal{NC}4$ | Avg. ↓ |
| Plastic | 15.10 | 0.498 | 0.515 | **0.428** | 1.422 | 74.00 |
| **Fixed ETF** | **12.62** | **0.393** | **0.490** | 0.468 | **0.316** | **65.10** |
| No $L_2$ norm | 12.74 | 0.579 | 0.538 | **0.349** | 1.339 | 68.93 |
| $L_2$ **norm** | 12.62 | **0.393** | **0.490** | 0.468 | **0.316** | **65.10** |

trainable projector and the other one uses a frozen simplex ETF projector. We summarize our findings in Table 4. Our results indicate that the fixed simplex ETF projector strengthens NC more than a regular plastic projector as evidenced by lower $\mathcal{NC}1$. Consequently, the ETF projector outperforms plastic projector in OOD detection by an absolute 8.9%.

We also evaluate the impact of $L_2$ normalization on the projector embeddings. We train two models in an identical setting, the only variable we change is the $L_2$ normalization. We observe that $L_2$ normalization achieves a lower $\mathcal{NC}1$ value (thereby strengthening NC) and 3.83% (absolute) lower OOD detection error than its counterpart. These results demonstrate that using $L_2$ normalization helps induce NC and thereby enhances OOD detection performance. Additional results are shown in Appendix H.

Table 5: **NECO Vs. Our Method.** Various DNNs e.g., VGG17, ResNet18, and ViT-T are trained on ImageNet-100 dataset (ID). Reported OOD detection error, $\mathcal{E}_{\text{DET}}$ (%) is averaged across eight OOD datasets.

| Method | Avg. $\mathcal{E}_{\text{DET}}$ (%) ↓ | | |
|---|---|---|---|
| | VGG17 | ResNet18 | ViT-T |
| NECO | 77.82 | 88.13 | 85.67 |
| **Ours** | **65.10** (-12.72) | **69.70** (-18.43) | **83.16** (-2.51) |

Table 6: **Projector Configuration.** VGG17 models with different ETF projector configurations are trained on ImageNet-100 (ID) dataset. $D$ and $W$ denote the depth and width of the projector, respectively. Reported $\mathcal{E}_{\text{GEN}}$ (%) and $\mathcal{E}_{\text{DET}}$ (%) are averaged over eight OOD datasets.

| Config. | $\mathcal{E}_{\text{ID}}$ ↓ | Neural Collapse ↓ | | | | $\mathcal{E}_{\text{GEN}}$ | $\mathcal{E}_{\text{DET}}$ |
|---|---|---|---|---|---|---|---|
| | | $\mathcal{NC}1$ | $\mathcal{NC}2$ | $\mathcal{NC}3$ | $\mathcal{NC}4$ | Avg. ↓ | Avg. ↓ |
| $D = 1$ | 12.86 | 0.375 | 0.649 | 0.500 | 1.157 | 45.37 | 87.37 |
| **D = 2** | **12.62** | 0.393 | 0.490 | 0.468 | 0.316 | **41.85** | **65.10** |
| $W = 2$ | 13.48 | 0.320 | 0.667 | 0.376 | 0.493 | 43.33 | 69.73 |

### 6.5. State-of-the-art Comparison

To put our work in context with respect to existing methods, we compare our method with NECO (Ammar et al., 2024), a state-of-the-art OOD detection method based on NC. Since NECO does not address OOD generalization, we restrict this comparison to OOD detection only. We train multiple DNN architectures on ImageNet-100 (ID) and evaluate their performance on eight OOD datasets. Remarkably, our method consistently outperforms NECO across all settings. As shown in Table 5, our approach reduces the average OOD detection error by an absolute margin of 12.72% for VGG17, 18.43% for ResNet18, and 2.51% for ViT-T, highlighting its superior effectiveness. Comprehensive results across OOD datasets are presented in Table 22.

### 6.6. Ablation Studies

**Projector Design Criteria.** Here we ask: *does a deeper or wider projector achieve higher performance?* Results are summarized in Table 6. We find that the projector with depth 2 performs better than shallower or wider projectors. Table 19 in the Appendix contains comprehensive results.

**Group Normalization Enhances Transfer.** While the impact of BN on NC has been studied in prior work (Pan & Cao, 2023; Ergen et al., 2022), we evaluate the effectiveness of both BN and GN within our framework. We compare BN with GN (GN is combined with WS) and show the results in Table 7. We find that GN helps mitigate NC in the encoder as indicated by a higher $\mathcal{NC}1$ value than BN. This implies that, unlike GN, BN leads to stronger NC and impairs OOD transfer. This is further confirmed by GN outperforming BN

Table 7: **Impact of Group Normalization.** Reported $\mathcal{E}_{\text{GEN}}$ (%) and $\mathcal{E}_{\text{DET}}$ (%) are averaged across eight OOD datasets. The comparison is based on ImageNet-100 pre-trained VGG17 networks. Reported $\mathcal{NC}1$ corresponds to the encoder.

| Method | $\mathcal{NC}1$ $\uparrow$ | $\mathcal{E}_{\text{ID}}$ $\downarrow$ | $\mathcal{E}_{\text{GEN}}$ Avg. $\downarrow$ | $\mathcal{E}_{\text{DET}}$ Avg. $\downarrow$ |
|---|---|---|---|---|
| Batch Normalization | 1.401 | 12.52 | 51.96 | 69.47 |
| **Group Normalization** | **2.175** | 12.62 | **41.85** | **65.10** |

Table 8: **SGD Optimizer.** VGG17 models are trained on ImageNet-100 (ID) dataset. Baseline VGG17 does not incorporate mechanisms like entropy regularization or the ETF projector to control NC. NC metrics are computed using the penultimate-layer embeddings. Reported $\mathcal{E}_{\text{GEN}}$ (%) and $\mathcal{E}_{\text{DET}}$ (%) are averaged over eight OOD datasets.

| Model | $\mathcal{E}_{\text{ID}}$ $\downarrow$ | Neural Collapse $\downarrow$ | | | | $\mathcal{E}_{\text{GEN}}$ Avg. $\downarrow$ | $\mathcal{E}_{\text{DET}}$ Avg. $\downarrow$ |
|---|---|---|---|---|---|---|---|
| | | $\mathcal{NC}1$ | $\mathcal{NC}2$ | $\mathcal{NC}3$ | $\mathcal{NC}4$ | | |
| VGG17 | 13.06 | 1.017 | 0.449 | 0.479 | 26.459 | 57.17 | 89.69 |
| **+Ours** | 13.18 | 0.087 | 0.468 | 0.267 | 0.264 | **50.91** | **60.81** |

by 10.11% (absolute) in OOD generalization. Our results suggest that replacing BN is crucial for OOD generalization. Furthermore, using GN improves OOD detection by 4.37% (absolute). Table 17 includes comprehensive results.

**SGD Optimizer.** While our main experiments employed the AdamW optimizer, we also evaluate the effectiveness of our method with the widely used SGD optimizer to ensure its robustness across optimization schemes. To this end, we train VGG17 models on ImageNet-100 dataset (ID) using SGD optimizer and assess their performance on eight OOD datasets. As shown in Table 8, our method outperforms the baseline by 6.26% (absolute) in OOD generalization and by 28.88% (absolute) in OOD detection. Comprehensive results are provided in Appendix H.6.

**Impact of Loss Functions: MSE Vs. CE.** Both MSE and CE are effective loss functions to achieve NC properties (Zhou et al., 2022). Unlike prior work, we evaluate their efficacy in both OOD detection and OOD generalization tasks. As shown in Table 9, CE outperforms MSE by 6.74% (absolute) in OOD detection and by 17.71% (absolute) in OOD generalization. Our observations are consistent with prior work (Kornblith et al., 2021; Hui & Belkin, 2020).

**Computational Efficiency.** Our method is computationally efficient, introducing minimal overhead compared to standard DNNs. We assess efficiency by measuring training time and FLOPs relative to baseline models. As shown in Table 23, the additional cost remains below 0.3% across all cases—a negligible overhead given the substantial performance gains. Further details are provided in Appendix H.8.

## 7. Discussion

Our study highlights the impact of neural collapse on OOD detection and OOD generalization. Several promising directions remain for future research. Extending our approach to open-world continual learning (Kim et al., 2025; Dong et al., 2024) presents an exciting challenge. While we focused on architectural and regularization-based techniques to control NC, another avenue is optimization-driven strategies. For instance, Markou et al. (2024) studies optimising towards the nearest simplex ETF to accelerate NC. Guiding NC to enhance task-specific representations or disentangle conflicting tasks could improve robustness and generalization. Moreover, beyond standard loss functions, alternative formulations could be explored to regulate NC.

Following prior work, our study primarily focused on vision tasks and datasets. However, extending our method to other modalities such as audio and text represents a promising direction for future research. While our experiments centered on classification tasks, the proposed method is inherently general and can be applied to other tasks, e.g., object detection or other regression tasks. The core of our approach—the fixed ETF projector—is designed to enforce NC in the final layer, enhancing feature representations that benefit both classification and regression tasks. Furthermore, our entropy regularization is task-agnostic and seamlessly integrates with both classification and regression objectives.

Our study utilized nearest neighbor density estimation for entropy regularization. Exploring parametric and adaptive approaches could offer more robust regularization techniques for improving OOD generalization. We demonstrated that controlling NC improves OOD detection and generalization, but a deeper theoretical understanding of this relationship is needed. Future work could establish theoretical frameworks that unify OOD detection and generalization from an NC perspective, offering a more comprehensive view of representation learning under distribution shifts.

## 8. Conclusion

In this work, we established a concrete relationship between neural collapse and OOD detection and generalization. Motivated by this relationship, our method enhances OOD detection by strengthening NC while promoting OOD generalization by mitigating NC. We also provided a theoretical framework to mitigate NC via entropy regularization. Our method demonstrated strong OOD detection and generalization abilities compared to baselines that did not control NC. This work has implications for open-world problems where simultaneous OOD detection and generalization are critical. We hope our work inspires future efforts to develop more effective methods for building robust AI systems in open-world conditions.

**Acknowledgments.** This work was partly supported by NSF awards #2326491, #2125362, and #2317706. The views and conclusions contained herein are those of the authors and should not be interpreted as representing any sponsor's official policies or endorsements. We thank Junyu Chen, Helia Dinh, and Shikhar Srivastava for their feedback and comments on the manuscript.

## Impact Statement

This paper aims to contribute to the advancement of the field of Machine Learning. While our work has the potential to influence various societal domains, we do not identify any specific societal impacts that require particular emphasis at this time.

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

# Appendix

We organize the Appendix as follows:

- Appendix A describes the implementation details. It describes the DNN architectures (VGG, ResNet, and ViT), feature extraction for linear probing, training, and evaluation details of both pre-training and linear probing in various experiments.

- Appendix B provides details on the datasets used in this paper. In total, we use 9 datasets.

- Appendix C describes four neural collapse metrics ($\mathcal{NC}1 - \mathcal{NC}4$) used in this paper.

- Appendix D presents a comprehensive comparison between MSE and CE.

- Appendix E contains proof on the implication of NC on entropy.

- Appendix F provides a comprehensive comparison between the encoder and projector across different architectures.

- Appendix G provides detailed analyses on entropy regularization and neural collapse.

- Additional experiments and analyses are summarized in Appendix H. The mechanisms of controlling NC have been examined.

- Appendix I includes the list of 100 classes in the ImageNet-100 dataset.

## A. Implementation Details

In this paper, we use several acronyms such as **NC** : Neural Collapse, **ETF** : Equiangular Tight Frame, **ID** : In-Distribution, **OOD** : Out-of-Distribution, **LR** : Learning Rate, **WD** : Weight Decay, **GAP** : Global Average Pooling, **GN** : Group Normalization, **BN** : Batch Normalization, **WS** : Weight Standardization, **CE** : Cross Entropy, **MSE** : Mean Squared Error, **FPR** : False Positive Rate.

**We use the terms "OOD generalization" and "OOD transfer" interchangeably.**

### A.1. Architectures

**VGG.** We modified the VGG-19 architecture to create our VGG-17 encoder. Additionally, we removed two fully connected (FC) layers before the final classifier head. And, we added an adaptive average pooling layer (nn.AdaptiveAvgPool2d), which allows the network to accept any input size while keeping the output dimensions the same. After VGG-17 encoder, we attached a projector

consisting of two MLP layers ($512 \rightarrow 2048 \rightarrow 512$) and finally added a classifier head. We use ReLU activation between projector layers. We replace BN with GN+WS in all layers. For GN, we use 32 groups in all layers.

**ResNet.** We used the entire ResNet-18 or ResNet-34 as the encoder and attached a projector ($512 \rightarrow 2048 \rightarrow 512$) similar to the VGG networks mentioned above. We replace BN with GN+WS in all layers. For GN, we use 32 groups in all layers.

**ViT.** We consider ViT-Tiny/Small (5.73M/21.85M parameters) as the encoder for our experiments. The projector comprising two MLP layers configured as fixed ETF Simplex and added after the encoder. Following (Beyer et al., 2022), we omit the learnable position embeddings and instead use the fixed 2D sin-cos position embeddings. Other details adhere to the original ViT paper (Dosovitskiy et al., 2020).

1. **ViT-Tiny Configuration:** patch size=16, embedding dimension=192, # heads=3, depth=12. Projector has output dimension=192 and hidden dimension=768, ($192 \rightarrow 768 \rightarrow 192$). We use ReLU activation between projector layers. The number of parameters in ViT-Tiny + projector is 6.02M.

2. **ViT-Small Configuration:** patch size=16, embedding dimension=384, # heads=6, depth=12. Projector has output dimension=384 and hidden dimension=1536, ($384 \rightarrow 1536 \rightarrow 384$). We use ReLU activation between projector layers. The number of parameters in ViT-Small + projector is 23.03M.

### A.2. Feature Extraction For Linear Probing

In experiments with CNNs, at each layer $l$, for each sample, we extract features of dimension $H_l \times W_l \times C_l$, where $H_l$, $W_l$, and $C_l$ denote the height, width and channel dimensions respectively. Next, following (Sarfi et al., 2023), we apply $2 \times 2$ adaptive average pooling on each spatial tensor ($H_l \times W_l$). After average pooling, features of dimension $2 \times 2 \times C_l$ are flattened and converted into a vector of dimension $4C_l$. Finally, a linear probe is trained on the flattened vectors. In experiments with ViTs, following (Raghu et al., 2021), we apply global average-pooling (GAP) to aggregate image tokens excluding the class token and train a linear probe on top of GAP tokens. We report the best error (%) on the test dataset for linear probing at each layer.

### A.3. VGG Experiments

**VGG ID Training:** For training VGG on ImageNet-100, we employ the AdamW optimizer with a LR of $6 \times 10^{-3}$ and WD of $5 \times 10^{-2}$ for batch size 512. The model is trained for 100 epochs using the Cosine Annealing LR scheduler with

a linear warmup of 5 epochs. In all experiments, we use CE and entropy regularization ($\alpha = 0.05$) losses. However, in some particular experiments comparing CE and MSE, we use MSE loss ($\kappa$=15, M=60) and entropy regularization loss ($\alpha = 0.05$). In the experiments with SGD optimizer and CE loss, we set LR to 0.2 and WD to $10^{-4}$ for batch size 512.

**VGG Linear Probing:** We use the AdamW optimizer with a flat LR of $1 \times 10^{-3}$ and WD of 0 for batch size 128. The linear probes are trained for 30 epochs. We use label smoothing of 0.1 with the cross-entropy loss.

### A.4. ResNet Experiments

**ResNet ID Training:** For training ResNet-18/34, we employ the AdamW optimizer with an LR of 0.01 and a WD of 0.05 for batch size 512. The model is trained for 100 epochs using the Cosine Annealing LR scheduler with a linear warmup of 5 epochs. We use CE and entropy regularization ($\alpha = 0.05$) losses.

**ResNet Linear Probing:** In the linear probing experiment, we use the AdamW optimizer with an LR of $1 \times 10^{-3}$ and WD of 0 for batch size 128. The linear probes are trained for 30 epochs. We use label smoothing of 0.1 with cross-entropy loss.

### A.5. ViT Experiments

**ViT ID Training:** For training ViT-Tiny, we employ the AdamW optimizer with LR of $8 \times 10^{-4}$ and WD of $5 \times 10^{-2}$ for batch size 256. The LR is scaled for $n$ GPUs according to: $LR \times n \times \frac{batchsize}{512}$. We use an LR of $4 \times 10^{-4}$ for ViT-Small when the batch size is 256. We use the Cosine Annealing LR scheduler with warm-up (5 epochs). We train the ViT-Tiny/Small for 100 epochs using CE and entropy regularization ($\alpha = 0.05$) losses. Following (Raghu et al., 2021; Beyer et al., 2022), we omit class token and instead use GAP token by global average-pooling image tokens and feed GAP embeddings to the projector.

**ViT Linear Probing:** We use the AdamW optimizer with LR of 0.01 and WD of $1 \times 10^{-4}$ for batch size 512. The linear probes are trained for 30 epochs. We use label smoothing of 0.1 with cross-entropy loss.

**Augmentation.** We use random resized crop and random flip augmentations and $224 \times 224$ images as inputs to the DNNs.

In experiments with CE loss, we use label smoothing of 0.1.

### A.6. Evaluation Criteria

**FPR95.** The OOD detection performance is evaluated by the FPR (False Positive Rate) metric. In particular, we use FPR95 (FPR at 95% True Positive Rate) that evaluates OOD detection performance by measuring the fraction of OOD samples misclassified as ID where threshold, $\lambda$ is chosen when the true positive rate is 95%. Both OOD detection and OOD generalization tasks are evaluated on the *same* OOD test set.

**Percentage Change.** To capture percentage increase or decrease when switching from the encoder ($E$) to the projector ($P$), we use

$$\Delta_{E \to P} = \frac{(P - E)}{|E|} \times 100.$$

**Normalization for different OOD datasets.** In our correlation analysis between NC and OOD detection/generalization (Fig. 1 and 7), we use min-max normalization for layer-wise OOD detection errors and OOD generalization errors which enables comparison using different OOD datasets. For a given OOD dataset and a DNN consisting of total $L$ layers, let the OOD detection/ generalization error for a layer $l$ be $E_l$. For $L$ layers we have error vector $\mathbf{E} = [E_1, E_2, \cdots E_L]$ which is then normalized by

$$\mathbf{E}_N = \frac{\mathbf{E} - \min(\mathbf{E})}{\max(\mathbf{E}) - \min(\mathbf{E})}.$$

**Effective Rank.** We use RankMe (Garrido et al., 2023) to measure the effective rank of the embeddings.

## B. Datasets

**ImageNet-100.** ImageNet-100 (Tian et al., 2020) is a subset of ImageNet-1K (Deng et al., 2009) and contains 100 ImageNet classes. It consists of 126689 training images ($224 \times 224$) and 5000 test images. The object categories present in ImageNet-100 are listed in Appendix I.

**CIFAR-100.** CIFAR-100 (Krizhevsky & Hinton, 2014) is a dataset widely used in computer vision. It contains $60,000$ RGB images and 100 classes, each containing 600 images. The dataset is split into $50,000$ training samples and $10,000$ test samples. The images in CIFAR-100 have a resolution of $32 \times 32$ pixels. Unlike CIFAR-10, CIFAR-100 has a higher level of granularity, with more fine-grained classes such as flowers, insects, household items, and a variety of animals and vehicles. For linear probing, all samples from both the training and validation datasets were used.

**NINCO (No ImageNet Class Objects).** NINCO (Bitterwolf et al., 2023) is a dataset with 64 classes. The dataset is curated to eliminate semantic overlap with ImageNet-1K dataset and is used to evaluate the OOD performance of the models pre-trained on imagenet-1K. The NINCO dataset has 5878 samples, and we split it into 4702 samples for training and 1176 samples for evaluation. We do not have a fixed number of samples per class for training and evaluation datasets.

**ImageNet-Rendition (ImageNet-R).** ImageNet-R incorporates distribution shifts using different artistic renditions of object classes from the original ImageNet dataset (Hendrycks et al., 2021). We use a variant of ImageNet-R dataset from (Wang et al., 2022b). ImageNet-R is a challenging benchmark for continual learning, transfer learning, and OOD detection. It consists of classes with different styles and intra-class diversity and thereby poses significant distribution shifts for ImageNet-1K pre-trained models (Wang et al., 2022b). It contains 200 classes, 24000 training images, and 6000 test images.

**CUB-200.** CUB-200 is composed of 200 different bird species (Wah et al., 2011). The CUB-200 dataset comprises a total of 11,788 images, with 5,994 images allocated for training and 5,794 images for testing.

**Aircrafts-100.** Aircrafts or FGVCAircrafts dataset (Maji et al., 2013) consists of 100 different aircraft categories and 10000 high-resolution images with 100 images per category. The training and test sets contain 6667 and 3333 images respectively.

**Oxford Pets-37.** The Oxford Pets dataset includes a total of 37 various pet categories, with an approximately equal number of images for dogs and cats, totaling around 200 images for each category (Parkhi et al., 2012).

**Flowers-102.** The Flowers-102 dataset contains 102 flower categories that can be easily found in the UK. Each category of the dataset contains 40 to 258 images. (Nilsback & Zisserman, 2008)

**STL-10.** STL-10 has 10 classes with 500 training images and 800 test images per class (Coates et al., 2011).

For all datasets, images are resized to $224 \times 224$ to train and evaluate DNNs.

## C. Neural Collapse Metrics

Neural Collapse (NC) describes a structured organization of representations in DNNs (Papyan et al., 2020; Kothapalli, 2023; Zhu et al., 2021; Rangamani et al., 2023). The following four criteria characterize Neural Collapse:

1. **Feature Collapse** ($\mathcal{NC}1$): Features within each class concentrate around a single mean, with almost no variability within classes.

2. **Simplex ETF Structure** ($\mathcal{NC}2$): Class means, when centered at the global mean, are linearly separable, maximally distant, and form a symmetrical structure on a hypersphere known as a Simplex Equiangular Tight Frame (Simplex ETF).

3. **Self-Duality** ($\mathcal{NC}3$): The last-layer classifiers align closely with their corresponding class means, forming

a self-dual configuration.

4. **Nearest Class Mean Decision** ($\mathcal{NC}4$): The classifier operates similarly to the nearest class-center (NCC) decision rule, assigning classes based on proximity to the class means.

Here, we describe each NC metric used in our results. Let $\boldsymbol{\mu}_G$ denote the global mean and $\boldsymbol{\mu}_c$ the $c$-th class mean of the features, $\{\mathbf{z}_{c,i}\}$ at layer $l$, defined as follows:

$$\boldsymbol{\mu}_G = \frac{1}{nC} \sum_{c=1}^{C} \sum_{i=1}^{n} \mathbf{z}_{c,i}, \quad \boldsymbol{\mu}_c = \frac{1}{n} \sum_{i=1}^{n} \mathbf{z}_{c,i} \quad (1 \le c \le C).$$

We drop the layer index $l$ from notation for simplicity. Also bias is excluded for notation simplicity. Feature dimension is $d$ instead of $d + 1$.

**Within-Class Variability Collapse** ($\mathcal{NC}1$)**:** It measures the relative size of the within-class covariance $\boldsymbol{\Sigma}_W$ with respect to the between-class covariance $\boldsymbol{\Sigma}_B$ of the DNN features:

$$\boldsymbol{\Sigma}_W = \frac{1}{nC} \sum_{c=1}^{C} \sum_{i=1}^{n} (\mathbf{z}_{c,i} - \boldsymbol{\mu}_c)(\mathbf{z}_{c,i} - \boldsymbol{\mu}_c)^\top \in \mathbb{R}^{d \times d},$$

$$\boldsymbol{\Sigma}_B = \frac{1}{C} \sum_{c=1}^{C} (\boldsymbol{\mu}_c - \boldsymbol{\mu}_G)(\boldsymbol{\mu}_c - \boldsymbol{\mu}_G)^\top \in \mathbb{R}^{d \times d}.$$

The $\mathcal{NC}1$ metric is defined as:

$$\mathcal{NC}1 = \frac{1}{C} \operatorname{trace}\left(\boldsymbol{\Sigma}_W \boldsymbol{\Sigma}_B^\dagger\right),$$

where $\boldsymbol{\Sigma}_B^\dagger$ is the pseudo-inverse of $\boldsymbol{\Sigma}_B$. **Note that $\mathcal{NC}1$ is the most dominant indicator of neural collapse.**

**Convergence to Simplex ETF** ($\mathcal{NC}2$)**:** It quantifies the $\ell_2$ distance between the normalized simplex ETF and the normalized $\mathbf{W}\mathbf{W}^\top$, as follows:

$$\mathcal{NC}2 := \left\| \frac{\mathbf{W}\mathbf{W}^\top}{\|\mathbf{W}\mathbf{W}^\top\|_F} - \frac{1}{\sqrt{C-1}} \left( \mathbf{I}_C - \frac{1}{C} \mathbf{1}_C \mathbf{1}_C^\top \right) \right\|_F,$$

where $\mathbf{W} \in \mathbb{R}^{C \times d}$ denotes the weight matrix of the learned classifier.

**Convergence to Self-Duality** ($\mathcal{NC}3$)**:** It measures the $\ell_2$ distance between the normalized simplex ETF and the normalized $\mathbf{W}\mathbf{Z}$:

$$\mathcal{NC}3 := \left\| \frac{\mathbf{W}\mathbf{Z}}{\|\mathbf{W}\mathbf{Z}\|_F} - \frac{1}{\sqrt{C-1}} \left( \mathbf{I}_C - \frac{1}{C} \mathbf{1}_C \mathbf{1}_C^\top \right) \right\|_F,$$

where $\mathbf{Z} = [\mathbf{z}_1 - \boldsymbol{\mu}_G \cdots \mathbf{z}_C - \boldsymbol{\mu}_G] \in \mathbb{R}^{d \times C}$ is the centered class-mean matrix.

**Simplification to NCC** ($\mathcal{NC}4$)**:** It measures the collapse of bias $\mathbf{b}$:

$$\mathcal{NC}4 := \|\mathbf{b} + \mathbf{W}\boldsymbol{\mu}_G\|_2.$$

Table 9: **Comparison between MSE and CE.** VGG17 networks are trained on **ImageNet-100** dataset (ID). For OOD generalization we report $\mathcal{E}_{\mathrm{GEN}}$ (%) whereas for OOD detection we report $\mathcal{E}_{\mathrm{DET}}$ (%), both are averaged over eight OOD datasets. All models incorporate entropy regularization and the ETF projector to control NC. **A lower $\mathcal{NC}$ indicates stronger neural collapse.** $+\Delta_{E\rightarrow P}$ and $-\Delta_{E\rightarrow P}$ indicate % increase and % decrease respectively, when changing from the encoder ($E$) to projector ($P$).

| Method | $\mathcal{E}_{\mathrm{ID}}$ $\downarrow$ | Neural Collapse | | | | $\mathcal{E}_{\mathrm{GEN}}$ Avg. $\downarrow$ | $\mathcal{E}_{\mathrm{DET}}$ Avg. $\downarrow$ |
| | | $\mathcal{NC}1$ | $\mathcal{NC}2$ | $\mathcal{NC}3$ | $\mathcal{NC}4$ | | |
| **CE Loss** | | | | | | | |
| Projector | **12.62** | 0.393 | 0.490 | 0.468 | 0.316 | 66.36 | **65.10** |
| Encoder | 15.52 | 2.175 | 0.603 | 0.616 | 5.364 | **41.85** | 87.62 |
| $\Delta_{E\rightarrow P}$ | -18.69 | -81.93 | -18.74 | -24.03 | -94.11 | +58.57 | -25.70 |
| **MSE Loss** | | | | | | | |
| Projector | **14.04** | 0.469 | 0.743 | 0.279 | 0.382 | 70.87 | **71.84** |
| Encoder | 14.74 | 2.267 | 0.843 | 0.673 | 10.773 | **59.56** | 88.88 |
| $\Delta_{E\rightarrow P}$ | -4.75 | -79.31 | -11.86 | -58.54 | -96.45 | +18.99 | -19.17 |

## D. Mean Squared Error vs. Cross-Entropy

Prior work (Kornblith et al., 2021) finds that MSE rivals CE in ID classification task but underperforms CE in OOD transfer. However, the comparison between CE and MSE in OOD detection task remains unexplored. In this work, we find that CE significantly outperforms MSE in both OOD transfer and OOD detection tasks. As shown in Table 9, MSE underperforms CE by 6.74% (absolute) in OOD detection and by 17.71% (absolute) in OOD generalization. Our OOD generalization results are consistent with Kornblith et al. (2021). CE also obtains lower ID error than MSE, thereby showing good overall performance.

In terms of inducing neural collapse, both MSE and CE are effective and achieve lower NC values (i.e., stronger NC). However, our results suggest that CE does a better job than MSE in enhancing NC without sacrificing OOD transfer. We find MSE to be sensitive to the hyperparameters. The comparison on all OOD datasets is shown in Table 10.

## E. Formal Proposition: Collapsing Implies Entropy $-\infty$

**Proposition E.1** (Entropy under Class–Conditional Collapse). *Consider a mixture of $K$ class–conditional densities $\{p_{\ell,k}(\mathbf{z};\epsilon)\}_{k=1}^{K}$ in $\mathbb{R}^{d_\ell}$ with mixture weights $\{\pi_k\}_{k=1}^{K}$. Suppose that for each $k$, the density $p_{\ell,k}(\mathbf{z};\epsilon)$ is a member of a family indexed by $\epsilon > 0$ that converges in the weak sense to a Dirac delta, i.e.,*

$$\lim_{\epsilon\to 0} p_{\ell,k}(\mathbf{z};\epsilon) = \delta(\mathbf{z} - \boldsymbol{\mu}_{\ell,k}).$$

*Then, the differential entropy of the mixture*

$$p_\ell(\mathbf{z};\epsilon) = \sum_{k=1}^{K} \pi_k\, p_{\ell,k}(\mathbf{z};\epsilon)$$

*diverges to $-\infty$ in the limit $\epsilon \to 0$, that is,*

$$\lim_{\epsilon\to 0} H\big(p_\ell(\mathbf{z};\epsilon)\big) = -\infty.$$

*Detailed Proof.* We begin by considering the mixture distribution

$$p_\ell(\mathbf{z};\epsilon) = \sum_{k=1}^{K} \pi_k\, p_{\ell,k}(\mathbf{z};\epsilon).$$

For each $k$, assume that the density $p_{\ell,k}(\mathbf{z};\epsilon)$ satisfies

$$\lim_{\epsilon\to 0} p_{\ell,k}(\mathbf{z};\epsilon) = \delta(\mathbf{z} - \boldsymbol{\mu}_{\ell,k}),$$

and, importantly, that its differential entropy diverges as

$$\lim_{\epsilon\to 0} H\big(p_{\ell,k}(\mathbf{z};\epsilon)\big) = -\infty.$$

A concrete example is when $p_{\ell,k}(\mathbf{z};\epsilon)$ is a Gaussian with covariance $\epsilon I$. In that case,

$$H\big(p_{\ell,k}(\mathbf{z};\epsilon)\big) = \frac{d_\ell}{2}\log(2\pi e\epsilon),$$

which clearly tends to $-\infty$ as $\epsilon \to 0$.

**Step 1: Introduce a Latent Class Variable.** Define a discrete random variable $K$ taking values in $\{1,\ldots,K\}$ with $\Pr(K = k) = \pi_k$. Then, the joint distribution of $(\mathbf{Z}, K)$ is given by

$$p(\mathbf{z}, k; \epsilon) = \pi_k\, p_{\ell,k}(\mathbf{z};\epsilon).$$

**Step 2: Apply the Chain Rule for Differential Entropy.** Using the chain rule for differential entropy, we have

$$H(\mathbf{Z}, K) = H(K) + H(\mathbf{Z} \mid K).$$

Here, the entropy of the discrete variable $K$ is

$$H(K) = -\sum_{k=1}^{K} \pi_k \log \pi_k,$$

which is finite since there are only finitely many classes. The conditional entropy is given by

$$H(\mathbf{Z} \mid K) = \sum_{k=1}^{K} \pi_k\, H\big(p_{\ell,k}(\mathbf{z};\epsilon)\big).$$

**Step 3: Relate the Entropy of the Mixture to the Conditional Entropies.** By a standard property of conditional entropy, we have

$$H(\mathbf{Z}) = H(\mathbf{Z}, K) - H(K \mid \mathbf{Z}) \leq H(\mathbf{Z}, K).$$

Thus, the entropy of the mixture satisfies

$$H\big(p_\ell(\mathbf{z}; \epsilon)\big) = H(\mathbf{Z}) \leq H(K) + \sum_{k=1}^{K} \pi_k \, H\big(p_{\ell,k}(\mathbf{z}; \epsilon)\big).$$

**Step 4: Conclude that the Mixture Entropy Diverges to** $-\infty$**.** Since $H(K)$ is a finite constant and for each $k$,

$$\lim_{\epsilon \to 0} H\big(p_{\ell,k}(\mathbf{z}; \epsilon)\big) = -\infty,$$

it follows that

$$\lim_{\epsilon \to 0} \sum_{k=1}^{K} \pi_k \, H\big(p_{\ell,k}(\mathbf{z}; \epsilon)\big) = -\infty.$$

Therefore,

$$\lim_{\epsilon \to 0} H\big(p_\ell(\mathbf{z}; \epsilon)\big) = -\infty.$$

**Discussion.** The essential idea is that even though the mixture might appear to smooth the singular behavior of individual class–conditional distributions, the overall entropy is still governed by the weighted sum of the entropies of its components. Because each component entropy diverges to $-\infty$, the entire mixture's entropy must also diverge to $-\infty$, up to the finite additive constant $H(K)$.

This completes the proof. $\square$

# F. Comprehensive Results (Encoder Vs. Projector)

## F.1. VGG Experiments

The detailed VGG17 results are given in Table 10. VGG results demonstrate that the encoder effectively mitigates NC for OOD generalization and the projector builds collapsed features and excels at the OOD detection task. The results also confirm that NC properties can be built using both CE and MSE loss functions.

**Qualitative Comparison.** We compare and visualize encoder embeddings and projector embeddings using UMAP. We also visualize the energy score distribution of ID and OOD data. The analysis is based on the VGG17 model pre-trained on the ImageNet-100 (ID) dataset and evaluated on OOD datasets: NINCO-64, Flowers-102, and STL-10. We observe the following:

- In Fig. 5a, the UMAP shows that projector embeddings nicely separate ID and OOD sets whereas encoder embeddings exhibit substantial overlap between ID and OOD

sets. This demonstrates that, unlike the encoder, the projector can intensify NC and is adept at OOD detection.
- We show the energy distribution of ID and OOD sets in Fig. 5b and 6. In all comparisons, we observe that the projector outperforms the encoder in separating ID and OOD sets based on energy scores.

## F.2. ResNet Experiments

The detailed ResNet18/34 results are given in Table 11. Our findings validate that NC can be controlled in various ResNet architectures for improving OOD detection and OOD generalization performance. Additionally, NC shows a strong correlation with OOD detection and OOD generalization as illustrated in Fig. 7.

We also visualize embeddings extracted from the encoder and projector of the ResNet18 model. As depicted in Fig. 8, projector embeddings exhibit much greater neural collapse than encoder embeddings.

## F.3. ViT Experiments

As shown in Table 12, the projector outperforms the encoder in OOD detection by absolute 7.73% (ViT-Tiny) and 9.23% (ViT-Small). Whereas the encoder outperforms the projector in OOD transfer by absolute 10.90% (ViT-Tiny) and 11.56% (ViT-Small). This demonstrates that controlling NC improves OOD detection and generalization in ViTs.

# G. Analysis on Entropy Regularization

Table 13 presents the detailed comparison between a model using the entropy regularization vs another model omitting it. We observe that using entropy penalty enhances OOD transfer by 2.71% (absolute), OOD detection by 2.36% (absolute), and ID performance by 0.84% (absolute).

Additionally, we analyze the impact of the entropy regularization loss coefficient on the ID and OOD transfer. Table 14 shows that increasing coefficient increases OOD transfer and rank of embeddings. This suggests that entropy regularization helps encode diverse features and reduce redundant features, encouraging utilization of all dimensions. Although entropy regularization is not sensitive to coefficient, over-regularization may hurt ID performance. Thereby, any non-aggressive coefficient can maintain good performance in both ID and OOD tasks.

We also analyze the impact of entropy regularization on encoder embeddings during the training phase. During each training epoch, we measure the NC1 criterion, entropy, and effective rank of encoder embeddings. These experiments are computationally intensive for large-scale datasets. Therefore, we perform small-scale experiments where we train VGG17 models on the ImageNet-10 (10 ImageNet classes)

Table 10: **Comprehensive VGG Results.** VGG17 models are trained on **ImageNet-100** dataset (ID) and evaluated on eight OOD datasets. All models incorporate entropy regularization and the ETF projector to control NC. For OOD transfer we report $\mathcal{E}_{\text{GEN}}$ (%) whereas for OOD detection we report $\mathcal{E}_{\text{DET}}$ (%). **A lower $\mathcal{NC}$ indicates stronger neural collapse.** The same color highlights the rows to compare.

| Method | $\mathcal{E}_{\text{ID}} \downarrow$ IN 100 | Neural Collapse $\mathcal{NC}1$ | $\mathcal{NC}2$ | $\mathcal{NC}3$ | $\mathcal{NC}4$ | OOD Datasets IN-R 200 | CIFAR 100 | Flowers 102 | NINCO 64 | CUB 200 | Aircrafts 100 | Pets 37 | STL 10 | Avg. |
|---|---|---|---|---|---|---|---|---|---|---|---|---|---|---|
| **CE Loss** | | | | | | | | | | | | | | |
| **Transfer Error ↓** | | | | | | | | | | | | | | |
| Projector | **12.62** | 0.393 | 0.490 | 0.468 | 0.316 | 91.38 | 65.72 | 64.51 | 64.97 | 82.22 | 97.42 | 43.17 | 21.51 | 66.36 |
| **Encoder** | 15.52 | 2.175 | 0.603 | 0.616 | 5.364 | **71.52** | **47.24** | **25.10** | **24.32** | **63.67** | **67.81** | **21.56** | **13.55** | **41.85** |
| **Detection Error ↓** | | | | | | | | | | | | | | |
| **Projector** | **12.62** | 0.393 | 0.490 | 0.468 | 0.316 | **60.85** | **48.23** | **42.35** | **67.69** | **56.51** | **99.04** | **76.32** | **69.84** | **65.10** |
| Encoder | 15.52 | 2.175 | 0.603 | 0.616 | 5.364 | 67.17 | 98.14 | 81.76 | 84.95 | 84.57 | 99.70 | 97.36 | 87.34 | 87.62 |
| **MSE Loss** | | | | | | | | | | | | | | |
| **Transfer Error ↓** | | | | | | | | | | | | | | |
| Projector | **14.04** | 0.469 | 0.743 | 0.279 | 0.382 | 87.18 | 70.33 | 82.16 | 55.95 | 90.35 | 97.09 | 55.93 | 28.01 | 70.87 |
| **Encoder** | 14.74 | 2.267 | 0.843 | 0.673 | 10.773 | **83.22** | **60.55** | **66.27** | **40.73** | **78.89** | **88.27** | **36.17** | **22.41** | **59.56** |
| **Detection Error ↓** | | | | | | | | | | | | | | |
| **Projector** | **14.04** | 0.469 | 0.743 | 0.279 | 0.382 | **63.75** | **48.02** | **61.18** | **69.50** | **74.58** | **99.10** | **84.57** | **74.06** | **71.84** |
| Encoder | 14.74 | 2.267 | 0.843 | 0.673 | 10.773 | 93.32 | 62.42 | 77.55 | 92.14 | 95.77 | 99.19 | 99.13 | 91.50 | 88.88 |

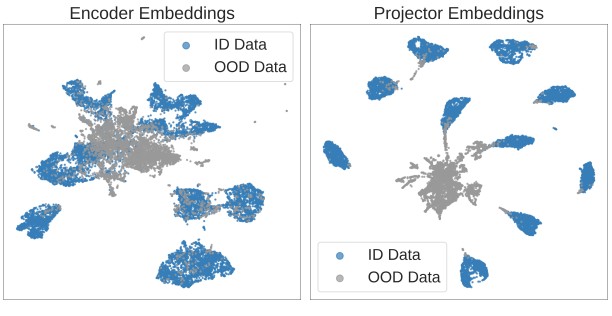

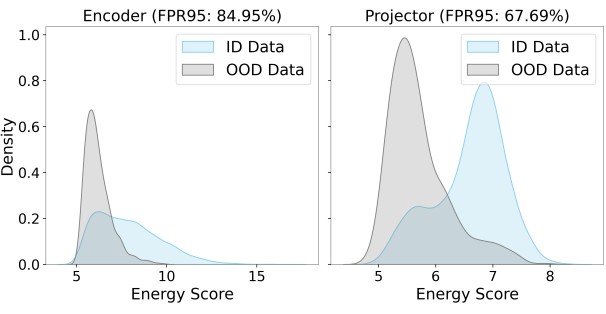

(a) UMAP of Embeddings

(b) Energy Score Distribution

Figure 5: **ID & OOD Data Visualization.** In **(a)**, The projector exhibits a greater separation between ID and OOD embeddings than the encoder. For clarity, we show 10 ImageNet classes as ID data and 64 classes from the NINCO dataset as OOD data. In **(b)**, The projector achieves higher energy scores (and lower FPR95) for ID data. For ID and OOD datasets, we show ImageNet-100 and NINCO-64 respectively.

subset for 100 epochs. We evaluate two cases: one with entropy regularization and another without entropy regularization.

The results are illustrated in Fig. 9. We find that entropy regularization achieves higher NC1 values during training compared to the model without any regularization. Thus, it helps mitigate NC during training, thereby contributing to OOD generalization. These findings align with our theoretical analysis showing entropy as an effective mechanism to prevent NC in the encoder.

Entropy regularization also increases the entropy and effective rank of the encoder embeddings. This demonstrates that entropy regularization helps encode diverse features, ensuring the features remain sufficiently "spread out."

Without the entropy regularization, the entropy of encoder embeddings does not improve. Also, the effective rank ends up at a low value (as low as the number of ID classes). The low rank is a sign of strong neural collapse and suggests that the encoder uses a few feature dimensions to encode information with huge redundancy in other dimensions. This degeneracy of embeddings impairs OOD transfer. Entropy regularization counteracts this and improves OOD transfer.

## H. Additional Experimental Results

### H.1. Fixed ETF Projector Vs. Learnable Projector

In Table 15, we observe that the fixed ETF projector shows a higher transfer error (2.47% absolute) than the plastic projector but outperforms the plastic projector in ID error

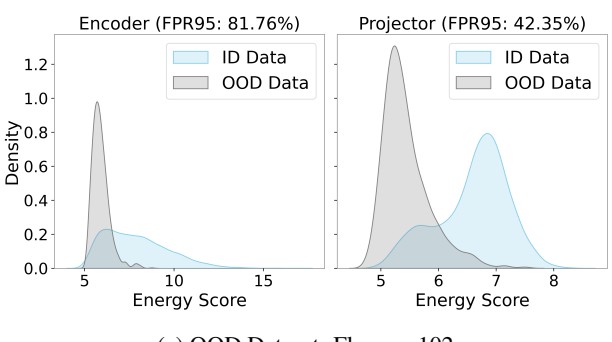 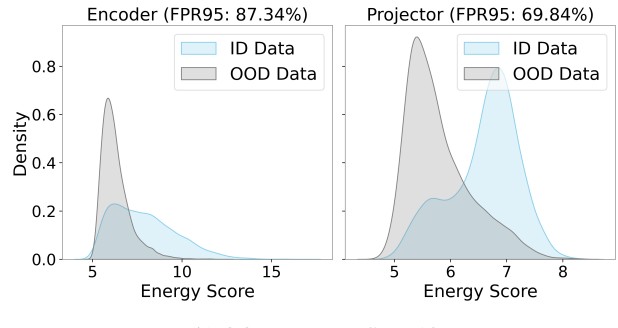

(a) OOD Dataset: Flowers-102          (b) OOD Dataset: STL-10

Figure 6: **Energy Score Distribution.** The projector creates a greater separation between ID and OOD data and achieves a lower FPR95 than the encoder. For better OOD detection, ID data should obtain higher energy scores than OOD data. For ID and OOD datasets, we show ImageNet-100 and Flowers-102/ STL-10 respectively. The energy scores are calculated based on logits from the VGG17 model pre-trained on ImageNet-100.

Table 11: **Comprehensive ResNet Results.** ResNet models are trained on **ImageNet-100** dataset (ID) and evaluated on eight OOD datasets. All models incorporate entropy regularization and the ETF projector to control NC. For OOD transfer we report $\mathcal{E}_{\text{GEN}}$ (%) whereas for OOD detection we report $\mathcal{E}_{\text{DET}}$ (%). **A lower $\mathcal{NC}$ indicates stronger neural collapse.**

| Model | $\mathcal{E}_{\text{ID}}\downarrow$ | Neural Collapse | | | | OOD Datasets | | | | | | | | |
|---|---|---|---|---|---|---|---|---|---|---|---|---|---|---|
| | IN 100 | $\mathcal{NC}1$ | $\mathcal{NC}2$ | $\mathcal{NC}3$ | $\mathcal{NC}4$ | IN-R 200 | CIFAR 100 | Flowers 102 | NINCO 64 | CUB 200 | Aircrafts 100 | Pets 37 | STL 10 | Avg. |
| **ResNet18** | | | | | | | | | | | | | | |
| **Transfer Error ↓** | | | | | | | | | | | | | | |
| Projector | **16.14** | 0.341 | 0.456 | 0.306 | 0.540 | 86.65 | 60.33 | 63.92 | 50.09 | 81.79 | 94.36 | 43.15 | 24.32 | 63.08 |
| **Encoder** | 20.14 | 1.762 | 0.552 | 0.555 | 10.695 | **74.17** | **53.33** | **31.37** | **28.15** | **68.85** | **81.61** | **27.72** | **16.56** | **47.72** |
| **Detection Error ↓** | | | | | | | | | | | | | | |
| **Projector** | **16.14** | 0.341 | 0.456 | 0.306 | 0.540 | **67.92** | **61.21** | **71.18** | **71.09** | **23.20** | **99.28** | **81.41** | **82.29** | **69.70** |
| Encoder | 20.14 | 1.762 | 0.552 | 0.555 | 10.695 | 71.50 | 96.44 | 86.27 | 84.78 | 65.48 | 99.43 | 95.86 | 89.63 | 86.17 |
| **ResNet34** | | | | | | | | | | | | | | |
| **Transfer Error ↓** | | | | | | | | | | | | | | |
| Projector | 14.54 | 0.252 | 0.672 | 0.294 | 0.324 | 83.93 | 58.65 | 64.41 | 44.05 | 81.65 | 93.58 | 43.64 | 22.87 | 61.60 |
| **Encoder** | 17.20 | 0.737 | 0.634 | 0.871 | 22.587 | **76.97** | **54.45** | **41.47** | **33.33** | **71.25** | **82.00** | **29.25** | **16.45** | **50.65** |
| **Detection Error ↓** | | | | | | | | | | | | | | |
| **Projector** | **14.54** | 0.252 | 0.672 | 0.294 | 0.324 | **61.72** | **60.05** | **47.94** | **66.24** | **67.59** | **98.35** | **83.78** | **78.49** | **70.52** |
| Encoder | 17.20 | 0.737 | 0.634 | 0.871 | 22.587 | 69.67 | 93.07 | 70.59 | 76.87 | 83.02 | 99.34 | 97.17 | 90.75 | 85.06 |

(2.48% absolute) and OOD detection error (8.9% absolute). A fixed ETF projector should intensify NC and hinder OOD transfer and our fixed ETF projector fulfills this goal.

## H.2. Impact of $L_2$ Normalization on NC

We verify whether $L_2$ normalization effectively induces more neural collapse and improves OOD detection. We analyze two VGG17 models pre-trained on ImageNet-100 dataset where one model uses $L_2$ normalization and the other omits it. The results are summarized in Table 16. We find that $L_2$ normalization induces more NC as evidenced by the lower NC1 value than its counterpart. Consequently, $L_2$ normalization improves OOD detection by 3.83% (absolute). Also, it achieves lower ID error than the compared model without $L_2$ normalization.

Next, we analyze how $L_2$ normalization impacts NC during

training. We perform small-scale experiments since large-scale experiments are compute-intensive. We train two VGG17 models on the ImageNet-10 (10 ImageNet classes) subset where one model uses $L_2$ normalization and another does not. During training, we measure the NC1 metric for the encoder embeddings. The impact of $L_2$ normalization on NC1 is exhibited in Fig. 9d. We find that $L_2$ normalization helps intensify NC during training. Consequently, it promotes better OOD detection.

## H.3. Batch Normalization Vs. Group Normalization

In this experiment, we analyze how batch normalization and group normalization perform within our framework. We find that group normalization (combined with weight standardization) outperforms batch normalization by 10.11% (absolute) in OOD generalization and by 4.37% (absolute) in

Table 12: **Comprehensive ViT Results.** ViT-Tiny (6.02M) and ViT-Small (21.62M) are trained on **ImageNet-100** dataset (ID) and evaluated on eight OOD datasets. All models incorporate entropy regularization and the ETF projector to control NC. For OOD transfer we report $\mathcal{E}_{\text{GEN}}$ (%) whereas for OOD detection we report $\mathcal{E}_{\text{DET}}$ (%). **A lower $\mathcal{NC}$ indicates stronger neural collapse.**

| Model | $\mathcal{E}_{\text{ID}} \downarrow$ IN 100 | Neural Collapse $\mathcal{NC}1$ | $\mathcal{NC}2$ | $\mathcal{NC}3$ | $\mathcal{NC}4$ | OOD Datasets IN-R 200 | CIFAR 100 | Flowers 102 | NINCO 64 | CUB 200 | Aircrafts 100 | Pets 37 | STL 10 | Avg. |
|---|---|---|---|---|---|---|---|---|---|---|---|---|---|---|
| **ViT-Tiny** | | | | | | | | | | | | | | |
| **Transfer Error ↓** | | | | | | | | | | | | | | |
| Projector | **32.04** | 2.748 | 0.609 | 0.798 | 1.144 | 87.37 | 60.71 | 64.61 | 39.71 | 80.00 | 92.00 | 54.27 | 29.55 | 63.53 |
| **Encoder** | 33.94 | 5.769 | 0.748 | 0.847 | 2.332 | **82.28** | **52.00** | **42.94** | **30.36** | **63.15** | **84.31** | **44.86** | **21.13** | **52.63** |
| **Detection Error ↓** | | | | | | | | | | | | | | |
| **Projector** | **32.04** | 2.748 | 0.609 | 0.798 | 1.144 | **81.12** | 60.81 | **77.55** | 82.40 | **79.05** | 99.10 | **95.15** | 90.06 | **83.16** |
| Encoder | 33.94 | 5.769 | 0.748 | 0.847 | 2.332 | 83.80 | 96.76 | 87.65 | 93.11 | 82.14 | 99.10 | 95.75 | **88.79** | 90.89 |
| **ViT-Small** | | | | | | | | | | | | | | |
| **Transfer Error ↓** | | | | | | | | | | | | | | |
| Projector | **31.28** | 0.822 | 0.522 | 0.712 | 0.962 | 86.57 | 58.46 | 64.51 | 39.20 | 78.25 | 90.70 | 53.86 | 29.30 | 62.61 |
| **Encoder** | 33.40 | 1.610 | 0.601 | 0.740 | 2.814 | **80.53** | **49.68** | **40.49** | **29.93** | **61.08** | **81.28** | **44.45** | **20.98** | **51.05** |
| **Detection Error ↓** | | | | | | | | | | | | | | |
| **Projector** | **31.28** | 0.822 | 0.522 | 0.712 | 0.962 | **76.03** | 58.79 | **75.20** | 81.97 | 82.46 | 98.50 | 95.42 | **88.74** | **82.14** |
| Encoder | 33.40 | 1.610 | 0.601 | 0.740 | 2.814 | 82.47 | 96.84 | 90.39 | 92.60 | 86.00 | 99.25 | **94.36** | 89.04 | 91.37 |

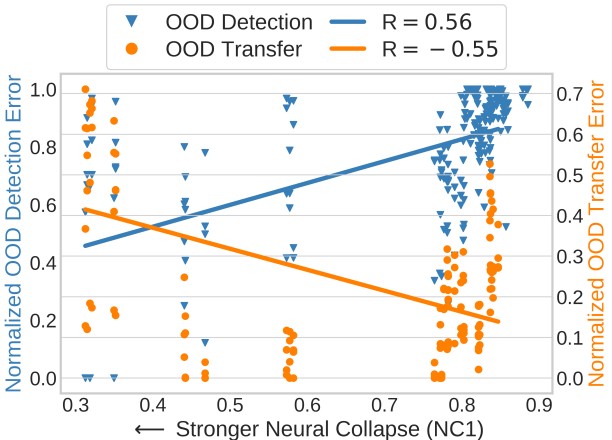

Figure 7: Lower NC1 values (indicating stronger neural collapse) correlate with lower OOD detection error but higher OOD transfer error, and vice versa. This suggests that stronger neural collapse improves OOD detection, while weaker neural collapse enhances OOD generalization. We analyze various layers of **ResNet18**, pre-trained on ImageNet-100 dataset (ID), and evaluate them on four OOD datasets. $R$ denotes the Pearson correlation coefficient.

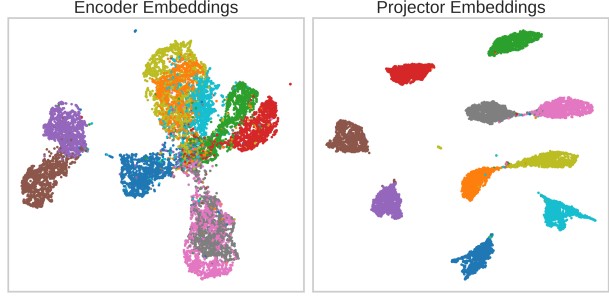

Figure 8: **Visualization of Embedding (ResNet18).** In this UMAP, projector embeddings exhibit greater neural collapse ($\mathcal{NC}1 = 0.341$) than the encoder embeddings ($\mathcal{NC}1 = 1.762$) as indicated by the formation of tight clusters around class-means. For clarity, we highlight 10 ImageNet classes by distinct colors. The embeddings are extracted from ImageNet-100 pre-trained ResNet18.

OOD detection (see Table 17). This demonstrates that batch normalization leads to less transferable representations.

Moreover, group normalization achieves a higher $\mathcal{NC}1$ value (i.e., lower neural collapse) than batch normalization, thereby mitigating NC effect and enhancing OOD generalization. Group normalization also achieves ID performance similar to that of batch normalization. Our results demonstrate that group normalization achieves competitive

performance and plays a crucial role in OOD generalization.

### H.4. Comparison with Baseline

Our experimental results show that our method significantly improves OOD detection and OOD transfer performance across all DNN architectures. We summarize the results in Table 18. We evaluate VGG17, ResNet18, and ViT-T baselines on 8 OOD datasets and compare them with our models. The absolute improvements over VGG17 baseline are 7.69% (OOD generalization) and 29.82% (OOD detection). Similarly, our method outperforms other DNNs in all criteria. Our results corroborate our argument that *controlling NC enables good OOD detection and OOD general-*

Table 13: **Entropy Regularization Vs. No Entropy Regularization.** VGG17 models are pre-trained on **ImageNet-100** dataset (ID) and evaluated on eight OOD datasets. **All models incorporate an ETF projector**. Entropy regularization loss with a coefficient, $\alpha$ is applied in the last encoder layer. The same color highlights the rows to compare. All metrics except NC are reported in %. The lower the NC value, the stronger the neural collapse. For OOD transfer we report $\mathcal{E}_{\text{GEN}}$ (%) whereas for OOD detection we report $\mathcal{E}_{\text{DET}}$ (%).

| Method | $\mathcal{E}_{\text{ID}} \downarrow$ | Neural Collapse | | | | OOD Datasets $\downarrow$ | | | | | | | | |
|---|---|---|---|---|---|---|---|---|---|---|---|---|---|---|
| | IN 100 | $\mathcal{NC}1$ | $\mathcal{NC}2$ | $\mathcal{NC}3$ | $\mathcal{NC}4$ | IN-R 200 | CIFAR 100 | Flowers 102 | NINCO 64 | CUB 200 | Aircrafts 100 | Pets 37 | STL 10 | Avg. |
| **Transfer Error $\downarrow$** | | | | | | | | | | | | | | |
| **No Reg. ($\alpha = 0$)** | | | | | | | | | | | | | | |
| Projector | 13.46 | 0.260 | 0.636 | 0.369 | 0.883 | 84.30 | 60.73 | 65.69 | 45.15 | 82.90 | 93.73 | 40.56 | 22.85 | 61.99 |
| Encoder | 15.24 | 1.308 | 0.719 | 0.619 | 5.184 | 73.52 | 49.26 | 37.06 | 25.51 | 64.31 | 69.58 | 22.24 | 15.00 | 44.56 |
| **Reg. ($\alpha = 0.05$)** | | | | | | | | | | | | | | |
| Projector | **12.62** | 0.393 | 0.490 | 0.468 | 0.316 | 91.38 | 65.72 | 64.51 | 64.97 | 82.22 | 97.42 | 43.17 | 21.51 | 66.36 |
| **Encoder** | 15.52 | 2.175 | 0.603 | 0.616 | 5.364 | **71.52** | **47.24** | **25.10** | 24.32 | 63.67 | **67.81** | **21.56** | **13.55** | **41.85** |
| **Reg. ($\alpha = 0.1$)** | | | | | | | | | | | | | | |
| Projector | 13.04 | 0.428 | 0.671 | 0.340 | 0.320 | 93.62 | 66.00 | 55.29 | 79.25 | 81.84 | 97.09 | 46.96 | 23.00 | 67.88 |
| Encoder | 16.12 | 2.861 | 0.538 | 0.636 | 6.677 | 73.05 | 48.61 | 27.84 | **22.62** | **61.91** | 70.21 | 22.87 | 13.83 | 42.62 |
| **Detection Error $\downarrow$** | | | | | | | | | | | | | | |
| **No Reg. ($\alpha = 0$)** | | | | | | | | | | | | | | |
| Projector | 13.46 | 0.260 | 0.636 | 0.369 | 0.883 | 65.22 | 54.32 | 45.20 | 67.18 | 52.37 | 98.41 | 84.38 | 72.58 | 67.46 |
| Encoder | 15.24 | 1.308 | 0.719 | 0.619 | 5.184 | 74.22 | 99.75 | 85.10 | 88.52 | 92.99 | 98.59 | 95.34 | 92.14 | 90.83 |
| **Reg. ($\alpha = 0.05$)** | | | | | | | | | | | | | | |
| **Projector** | **12.62** | 0.393 | 0.490 | 0.468 | 0.316 | **60.85** | **48.23** | **42.35** | 67.69 | 56.51 | 99.04 | **76.32** | **69.84** | **65.10** |
| Encoder | 15.52 | 2.175 | 0.603 | 0.616 | 5.364 | 67.17 | 98.14 | 81.76 | 84.95 | 84.57 | 99.70 | 97.36 | 87.34 | 87.62 |
| **Reg. ($\alpha = 0.1$)** | | | | | | | | | | | | | | |
| Projector | 13.04 | 0.428 | 0.671 | 0.340 | 0.320 | 61.13 | 54.69 | 43.14 | **64.63** | **50.73** | 98.74 | 82.42 | 71.51 | 65.87 |
| Encoder | 16.12 | 2.861 | 0.538 | 0.636 | 6.677 | 68.72 | 94.67 | 85.78 | 87.76 | 85.49 | 98.92 | 95.15 | 86.28 | 87.85 |

*ization performance*. It is also evident that a single feature space cannot simultaneously achieve both OOD detection and OOD generalization abilities.

## H.5. Projector Design Criteria

Here we study the design choices of the projector network. We want to know how depth and width impact the performance. For this, we examine projectors consisting of a single layer (*depth=1*, 512d), two layers (*depth=2*, $512d \rightarrow 2048d \rightarrow 512d$), three layers (*depth=3*, $512d \rightarrow 2048d \rightarrow 2048d \rightarrow 512d$), and a wider variant (*width=2*, $512d \rightarrow 4096d \rightarrow 512d$). All of these variants are trained in identical settings and only the projector is changed. We train VGG17 networks on ImageNet-100 dataset (ID) and evaluate OOD detection/generalization on 8 OOD datasets. The results are shown in Table 19. The projector with depth 2 outperforms other variants across all evaluations.

## H.6. SGD Optimizer

In our experiments, we mainly used AdamW optimizer and thereby we want to verify if our method works well with other commonly used optimizers e.g., SGD. For this, we train VGG17 models on ImageNet-100 dataset (ID) with the SGD optimizer and evaluate them on eight OOD datasets. As shown in Table 20, our method outperforms the baseline by an absolute 6.26% in OOD generalization and by an

absolute 28.88% in OOD detection. Also, we observe that our encoder reduces NC and enhances OOD generalization by an absolute 13.86% compared to the projector. Whereas the projector intensifies NC and improves OOD detection by an absolute 25.34% compared to the encoder. While SGD intensifies NC more than AdamW, AdamW achieves better overall performance (Table 18 in Appendix).

## H.7. Fixed ETF Classifier Vs. Plastic Classifier

We investigate how using a fixed ETF classifier head impacts NC and OOD detection/generalization performance. We train two identical models consisting of our proposed mechanisms to control NC, the only thing we vary is the classifier head. One model consists of a plastic (learnable) classifier head which is our proposed model and the other consists of an ETF classifier head. The ETF classifier head is configured with Simplex ETF and frozen during training. We train VGG17 networks on ImageNet-100 (ID) and evaluate them on 8 OOD datasets.

Table 21 shows results across all OOD datasets, where the plastic classifier outperforms the fixed ETF classifier by 4.39% (absolute) in OOD detection and by 15.6% in OOD generalization. The plastic classifier also outperforms ETF classifier in the ID task. Our results suggest that imposing NC in the classifier head is sub-optimal for enhancing OOD detection and generalization.

Table 14: **Entropy Regularization Loss Coefficient.** We examine the impact of entropy regularization on the OOD generalization of a regular **VGG17 without any projector**. VGG17 models are pre-trained on the ImageNet-10 (10 ImageNet classes) ID dataset and evaluated on eight OOD datasets. $\alpha$ denotes the entropy regularization loss coefficient. Effective rank corresponds to the penultimate embeddings where entropy regularization is applied or omitted. For OOD generalization, we report $\mathcal{E}_{\text{GEN}}$ (%).

| Reg. Coeff. | $\mathcal{E}_{\text{ID}} \downarrow$ | Rank $\uparrow$ | $\mathcal{E}_{\text{GEN}} \downarrow$ | | | | | | | | |
|---|---|---|---|---|---|---|---|---|---|---|---|
| $\alpha$ | IN | IN | IN-R | CIFAR | Flowers | NINCO | CUB | Aircrafts | Pets | STL | Avg. |
| | 10 | 10 | 200 | 100 | 102 | 64 | 200 | 100 | 37 | 10 | |
| 0 | **9.20** | 2211.99 | 94.62 | 83.77 | 72.45 | 65.56 | 86.76 | 89.32 | 80.32 | 49.44 | 77.78 |
| 0.1 | 9.80 | 2964.39 | 90.72 | 75.58 | 57.94 | 50.09 | 79.96 | 84.13 | 72.72 | 39.79 | 68.87 |
| 0.2 | 10.20 | 3170.92 | 90.25 | 72.77 | 57.84 | 50.85 | 79.48 | 84.16 | 70.43 | **37.71** | 67.94 |
| 0.6 | 12.00 | 3761.33 | **88.33** | 68.73 | **50.29** | 47.45 | **77.10** | 82.57 | **67.73** | 39.00 | 65.15 |
| 1.0 | 12.80 | **4815.32** | 88.38 | **67.81** | **50.29** | **47.11** | 77.48 | **81.64** | 68.27 | 38.74 | **64.96** |

Table 15: **ETF Fixed Projector Vs. Plastic Projector.** VGG17 models are trained on **ImageNet-100** dataset (ID) and evaluated on eight OOD datasets. All models incorporate entropy regularization and a projector (plastic/fixed ETF). The same color highlights the rows to compare. For OOD transfer we report $\mathcal{E}_{\text{GEN}}$ (%) whereas for OOD detection we report $\mathcal{E}_{\text{DET}}$ (%).

| Projector | $\mathcal{E}_{\text{ID}} \downarrow$ | Neural Collapse | | | | OOD Datasets | | | | | | | | |
|---|---|---|---|---|---|---|---|---|---|---|---|---|---|---|
| | IN | $\mathcal{NC}1$ | $\mathcal{NC}2$ | $\mathcal{NC}3$ | $\mathcal{NC}4$ | IN-R | CIFAR | Flowers | NINCO | CUB | Aircrafts | Pets | STL | Avg. |
| | 100 | | | | | 200 | 100 | 102 | 64 | 200 | 100 | 37 | 10 | |
| **Transfer Error** $\downarrow$ | | | | | | | | | | | | | | |
| **Plastic** | | | | | | | | | | | | | | |
| Projector | 15.10 | 0.498 | 0.515 | 0.428 | 1.422 | 87.52 | 64.83 | 79.71 | 53.32 | 87.00 | 93.46 | 48.76 | 28.04 | 67.83 |
| Encoder | 23.64 | 13.953 | 0.526 | 0.833 | 6.697 | **69.43** | **45.12** | **20.00** | **23.55** | **57.90** | **60.10** | 25.40 | 13.52 | **39.38** |
| **Fixed ETF (Ours)** | | | | | | | | | | | | | | |
| Projector | **12.62** | 0.393 | 0.490 | 0.468 | 0.316 | 91.38 | 65.72 | 64.51 | 64.97 | 82.22 | 97.42 | 43.17 | 21.51 | 66.36 |
| **Encoder** | 15.52 | 2.175 | 0.603 | 0.616 | 5.364 | 71.52 | 47.24 | 25.10 | 24.32 | 63.67 | 67.81 | **21.56** | 13.55 | 41.85 |
| **Detection Error** $\downarrow$ | | | | | | | | | | | | | | |
| **Plastic** | | | | | | | | | | | | | | |
| Projector | 15.10 | 0.498 | 0.515 | 0.428 | 1.422 | 63.05 | **47.87** | 62.45 | 70.07 | 80.88 | 98.95 | 89.37 | 79.25 | 74.00 |
| Encoder | 23.64 | 13.953 | 0.526 | 0.833 | 6.697 | 81.27 | 98.82 | 93.33 | 86.48 | 79.98 | 99.40 | 91.25 | 93.88 | 90.55 |
| **Fixed ETF (Ours)** | | | | | | | | | | | | | | |
| **Projector** | **12.62** | 0.393 | 0.490 | 0.468 | 0.316 | **60.85** | 48.23 | **42.35** | 67.69 | 56.51 | 99.04 | **76.32** | 69.84 | **65.10** |
| Encoder | 15.52 | 2.175 | 0.603 | 0.616 | 5.364 | 67.17 | 98.14 | 81.76 | 84.95 | 84.57 | 99.70 | 97.36 | 87.34 | 87.62 |

## H.8. Computational Efficiency & Scalability

Our proposed method is computationally efficient and does not require higher computational costs than standard DNNs. It introduces two additional components compared to standard DNN architecture and training protocol:

1. Entropy regularization applied at the encoder's output.
2. A frozen ETF projector (two MLP layers) following the DNN backbone.

For entropy regularization, we employ an efficient batch-level nearest neighbor distance computation, which incurs negligible computational overhead during training. Regarding the ETF projector, since it remains frozen and does not undergo gradient updates, it does not introduce any noticeable training costs beyond those of the baseline DNN.

**Training Time.** When training DNNs on ImageNet-100 (ID dataset) for 100 epochs using four NVIDIA RTX A5000 GPUs, both our method and the baseline require almost the same training time (see Table 23).

**FLOPs.** In terms of FLOPs (floating-point operations per second), both our method and the baseline require almost the same amount of computation. For FLOPs analysis, we use DeepSpeed [2] with the same GPU (single NVIDIA RTX A5000) across compared models. As shown in Table 23, the overhead introduced by our method remains below 0.3% in all cases, which we believe is trivial and well-justified given the observed performance gains.

**Scalability.** Here we ask: *does the proposed method scale to deeper architectures?* Our method is inherently compatible with deeper architectures since the ETF projector (two MLP layers) can be seamlessly integrated into encoders of any depth. Additionally, while deeper DNNs typically exhibit stronger NC in their top layers, our entropy regularizer effectively mitigates NC in encoders of any depth. As shown

[2] https://github.com/deepspeedai/DeepSpeed

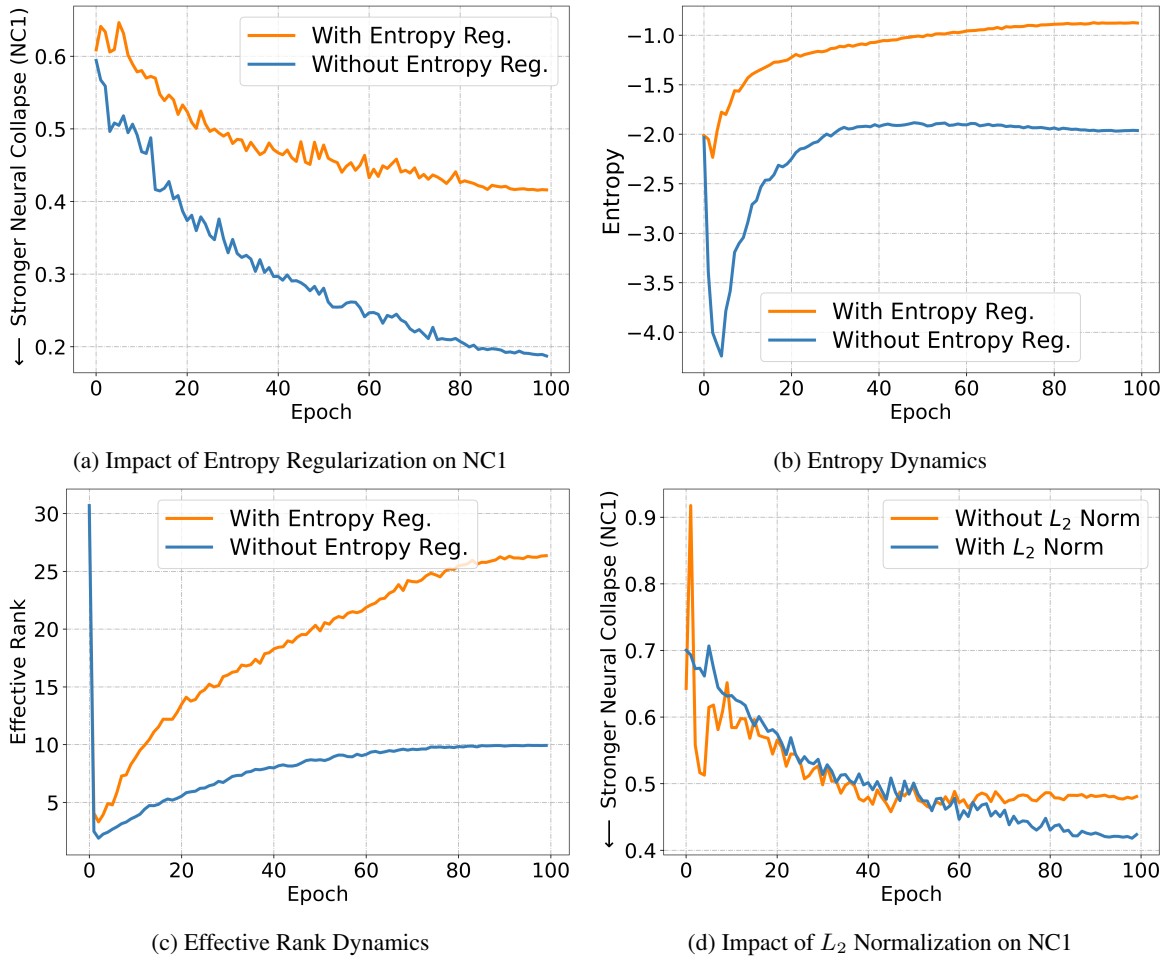

(a) Impact of Entropy Regularization on NC1

(b) Entropy Dynamics

(c) Effective Rank Dynamics

(d) Impact of $L_2$ Normalization on NC1

Figure 9: **Analyzing entropy regularization & $L_2$ normalization. (a)** Entropy regularization reduces neural collapse (indicated by higher NC1 values) in the encoder. **(b)** Entropy regularization increases the entropy of encoder embeddings otherwise entropy remains unchanged. **(c)** Entropy regularization increases the effective rank of encoder embeddings otherwise effective rank remains as low as the number of classes (i.e., 10 ImageNet classes). **(d)** $L_2$ normalization increases neural collapse (indicated by lower NC1 values) in the projector. For this analysis, we train VGG17 networks on the ImageNet-10 subset (10 ImageNet classes) for 100 epochs.

in Table 11, our method performs effectively with both ResNet18 and ResNet34, highlighting its scalability. Similar trend is observed for ViTs (Table 12).

## I. Classes of ImageNet-100 ID Dataset

We list the 100 classes in the ID dataset, ImageNet-100 (Tian et al., 2020). This list can also be found at: https://github.com/HobbitLong/CMC/blob/master/imagenet100.txt

*Rocking chair, pirate, computer keyboard, Rottweiler, Great Dane, tile roof, harmonica, langur, Gila monster, hognose snake, vacuum, Doberman, laptop, gasmask, mixing bowl, robin, throne, chime, bonnet, komondor, jean, moped, tub, rotisserie, African hunting dog, kuvasz, stretcher, garden spider, theater curtain, honeycomb, garter snake, wild boar, pedestal, bassinet, pickup, American lobster, sarong, mousetrap, coyote, hard disc, chocolate sauce, slide rule, wing, cauliflower, American Staffordshire terrier, meerkat, Chihuahua, lorikeet, bannister, tripod, head cabbage, stinkhorn, rock crab, papillon, park bench, reel, toy terrier, obelisk, walking stick, cocktail shaker, standard poodle, cinema, carbonara, red fox, little blue heron, gyromitra, Dutch oven, hare, dung beetle, iron, bottlecap, lampshade, mortarboard, purse, boathouse, ambulance, milk can, Mexican hairless, goose, boxer, gibbon, football helmet, car wheel, Shih-Tzu, Saluki, window screen, English foxhound, American coot, Walker hound, modem, vizsla, green mamba, pineapple, safety pin, borzoi, tabby, fiddler crab, leafhopper, Chesapeake Bay retriever, and ski mask.*

Table 16: **L₂ Normalization.** VGG17 models are trained on **ImageNet-100** dataset (ID) and evaluated on eight OOD datasets. All models incorporate entropy regularization and the ETF projector to control NC. The same color highlights the rows to compare. For OOD detection, we report $\mathcal{E}_{\text{DET}}$ (%).

| Method | $\mathcal{E}_{\text{ID}}\downarrow$ IN 100 | Neural Collapse $\downarrow$ $\mathcal{NC}1$ | $\mathcal{NC}2$ | $\mathcal{NC}3$ | $\mathcal{NC}4$ | $\mathcal{E}_{\text{DET}}\downarrow$ IN-R 200 | CIFAR 100 | Flowers 102 | NINCO 64 | CUB 200 | Aircrafts 100 | Pets 37 | STL 10 | Avg. |
|---|---|---|---|---|---|---|---|---|---|---|---|---|---|---|
| **No $L_2$ Norm** | | | | | | | | | | | | | | |
| Projector | 12.74 | 0.579 | 0.538 | **0.349** | 1.339 | **57.43** | 49.41 | 62.35 | 69.81 | 58.04 | 99.58 | 85.28 | 69.53 | 68.93 |
| Encoder | 14.70 | 1.788 | 0.633 | 0.823 | 10.643 | 77.08 | 96.77 | 91.18 | 92.35 | 89.47 | 99.64 | 89.51 | 85.31 | 90.16 |
| **$L_2$ Norm** | | | | | | | | | | | | | | |
| **Projector** | **12.62** | **0.393** | **0.490** | 0.468 | **0.316** | 60.85 | **48.23** | 42.35 | **67.69** | **56.51** | 99.04 | **76.32** | 69.84 | **65.10** |
| Encoder | 15.52 | 2.175 | 0.603 | 0.616 | 5.364 | 67.17 | 98.14 | 81.76 | 84.95 | 84.57 | 99.70 | 97.36 | 87.34 | 87.62 |

Table 17: **Batch Norm Vs. Group Norm.** VGG17 models are trained on **ImageNet-100** dataset (ID) and evaluated on eight OOD datasets. All models incorporate entropy regularization and the ETF projector to control NC. The same color highlights the rows to compare. Group norm is integrated with weight standardization. All metrics except NC are reported in percentage. For OOD transfer we report $\mathcal{E}_{\text{GEN}}$ (%) whereas for OOD detection we report $\mathcal{E}_{\text{DET}}$ (%).

| Method | $\mathcal{E}_{\text{ID}}\downarrow$ IN 100 | Neural Collapse $\mathcal{NC}1$ | $\mathcal{NC}2$ | $\mathcal{NC}3$ | $\mathcal{NC}4$ | OOD Datasets IN-R 200 | CIFAR 100 | Flowers 102 | NINCO 64 | CUB 200 | Aircrafts 100 | Pets 37 | STL 10 | Avg. |
|---|---|---|---|---|---|---|---|---|---|---|---|---|---|---|
| **Transfer Error $\downarrow$** | | | | | | | | | | | | | | |
| **Batch Norm** | | | | | | | | | | | | | | |
| Projector | 12.52 | 0.372 | 0.669 | 0.263 | 0.536 | 89.43 | 66.00 | 63.14 | 64.46 | 83.00 | 94.57 | 38.65 | 21.30 | 65.07 |
| Encoder | 14.54 | 1.401 | 0.605 | 0.590 | 25.611 | 78.02 | 53.34 | 49.51 | 33.25 | 74.08 | 85.27 | 25.46 | 16.75 | 51.96 |
| **Group Norm** | | | | | | | | | | | | | | |
| Projector | 12.62 | 0.393 | 0.490 | 0.468 | 0.316 | 91.38 | 65.72 | 64.51 | 64.97 | 82.22 | 97.42 | 43.17 | 21.51 | 66.36 |
| **Encoder** | 15.52 | 2.175 | 0.603 | 0.616 | 5.364 | **71.52** | **47.24** | **25.10** | **24.32** | **63.67** | **67.81** | **21.56** | **13.55** | **41.85** |
| **Detection Error $\downarrow$** | | | | | | | | | | | | | | |
| **Batch Norm** | | | | | | | | | | | | | | |
| Projector | 12.52 | 0.372 | 0.669 | 0.263 | 0.536 | **57.30** | 74.62 | 44.12 | **66.33** | 65.14 | 99.19 | 75.93 | 73.13 | 69.47 |
| Encoder | 14.54 | 1.401 | 0.605 | 0.590 | 25.611 | 92.17 | 99.77 | 91.08 | 91.41 | 99.48 | 98.62 | 85.39 | 93.26 | 93.90 |
| **Group Norm** | | | | | | | | | | | | | | |
| **Projector** | 12.62 | 0.393 | 0.490 | 0.468 | 0.316 | 60.85 | **48.23** | **42.35** | 67.69 | **56.51** | 99.04 | 76.32 | **69.84** | **65.10** |
| Encoder | 15.52 | 2.175 | 0.603 | 0.616 | 5.364 | 67.17 | 98.14 | 81.76 | 84.95 | 84.57 | 99.70 | 97.36 | 87.34 | 87.62 |

Table 18: **Comprehensive Comparison with Baseline.** Various DNNs are trained on **ImageNet-100** dataset (ID) and evaluated on eight OOD datasets. Baseline models do not incorporate mechanisms like entropy regularization or the ETF projector to control NC. NC metrics are computed using the penultimate-layer embeddings. For OOD transfer we report $\mathcal{E}_{\text{GEN}}$ (%) whereas for OOD detection we report $\mathcal{E}_{\text{DET}}$ (%).

| Model | $\mathcal{E}_{\text{ID}}\downarrow$ IN 100 | Neural Collapse $\mathcal{NC}1$ | $\mathcal{NC}2$ | $\mathcal{NC}3$ | $\mathcal{NC}4$ | OOD Datasets IN-R 200 | CIFAR 100 | Flowers 102 | NINCO 64 | CUB 200 | Aircrafts 100 | Pets 37 | STL 10 | Avg. |
|---|---|---|---|---|---|---|---|---|---|---|---|---|---|---|
| **Transfer Error $\downarrow$** | | | | | | | | | | | | | | |
| VGG17 | 12.18 | 0.766 | 0.705 | 0.486 | 37.491 | 75.60 | 50.11 | 42.75 | 29.17 | 71.35 | 84.13 | 27.58 | 15.65 | 49.54 |
| **VGG17+Ours** | 12.62 | 0.393 | 0.490 | 0.468 | 0.316 | **71.52** | **47.24** | **25.10** | **24.32** | **63.67** | **67.81** | **21.56** | **13.55** | **41.85** |
| **Detection Error $\downarrow$** | | | | | | | | | | | | | | |
| VGG17 | 12.18 | 0.766 | 0.705 | 0.486 | 37.491 | 96.02 | 97.16 | 97.94 | 93.11 | 95.19 | 98.59 | 87.33 | 94.05 | 94.92 |
| **VGG17+Ours** | 12.62 | 0.393 | 0.490 | 0.468 | 0.316 | **60.85** | **48.23** | **42.35** | **67.69** | **56.51** | 99.04 | **76.32** | **69.84** | **65.10** |
| **Transfer Error $\downarrow$** | | | | | | | | | | | | | | |
| ResNet18 | 15.38 | 1.11 | 0.658 | 0.590 | 31.446 | 75.75 | **49.48** | 41.37 | 30.02 | 69.80 | 82.75 | 29.63 | 16.53 | 49.42 |
| **ResNet18+Ours** | 16.14 | 0.341 | 0.456 | 0.306 | 0.540 | **74.17** | 53.33 | **31.37** | **28.15** | **68.85** | 81.61 | **27.72** | 16.56 | **47.72** |
| **Detection Error $\downarrow$** | | | | | | | | | | | | | | |
| ResNet18 | 15.38 | 1.11 | 0.658 | 0.590 | 31.446 | 98.40 | 98.85 | 98.33 | 96.68 | 96.60 | 99.67 | 92.40 | 98.25 | 97.40 |
| **ResNet18+Ours** | 16.14 | 0.341 | 0.456 | 0.306 | 0.540 | **67.92** | **61.21** | **71.18** | **71.09** | **23.20** | 99.28 | **81.41** | **82.29** | **69.70** |
| **Transfer Error $\downarrow$** | | | | | | | | | | | | | | |
| ViT-T | 31.78 | 2.467 | 0.657 | 0.601 | 1.015 | 82.18 | 52.64 | 41.67 | 32.74 | 63.48 | 81.61 | 45.11 | 22.00 | 52.68 |
| **ViT-T+Ours** | 32.04 | 2.748 | 0.609 | 0.798 | 1.144 | 82.28 | **52.00** | 42.94 | **30.36** | **63.15** | 84.31 | **44.86** | 21.13 | 52.63 |
| **Detection Error $\downarrow$** | | | | | | | | | | | | | | |
| ViT-T | 31.78 | 2.467 | 0.657 | 0.601 | 1.015 | 85.18 | 91.70 | 87.06 | 89.54 | 87.78 | **98.35** | 91.77 | 89.99 | 90.17 |
| **ViT-T+Ours** | 32.04 | 2.748 | 0.609 | 0.798 | 1.144 | **81.12** | **60.81** | **77.55** | **82.40** | **79.05** | 99.10 | **95.15** | 90.06 | **83.16** |

Table 19: **Projector Design Criteria.** VGG17 models are trained on **ImageNet-100** dataset (ID) and evaluated on eight OOD datasets. **All compared projectors are configured as fixed simplex ETFs**. And, entropy regularization is used in all cases. The same color highlights the rows to compare. Our final model has depth 2 and performs better than other variants. For OOD transfer we report $\mathcal{E}_{\text{GEN}}$ (%) whereas for OOD detection we report $\mathcal{E}_{\text{DET}}$ (%).

| Criteria | $\mathcal{E}_{\text{ID}}$↓ IN 100 | Neural Collapse | | | | OOD Datasets | | | | | | | | |
|---|---|---|---|---|---|---|---|---|---|---|---|---|---|---|
| | | $\mathcal{NC}1$ | $\mathcal{NC}2$ | $\mathcal{NC}3$ | $\mathcal{NC}4$ | IN-R 200 | CIFAR 100 | Flowers 102 | NINCO 64 | CUB 200 | Aircrafts 100 | Pets 37 | STL 10 | Avg. |
| **Transfer Error ↓** | | | | | | | | | | | | | | |
| **Depth=1** | | | | | | | | | | | | | | |
| Projector | 12.86 | 0.375 | 0.649 | 0.500 | 1.157 | 90.27 | 64.61 | 60.88 | 55.02 | 81.12 | 96.34 | 44.34 | 23.04 | 64.45 |
| Encoder | 16.34 | 1.673 | 0.667 | 0.589 | 7.936 | 74.08 | 50.61 | 30.00 | 28.06 | 66.74 | 71.95 | 25.73 | 15.75 | 45.37 |
| **Depth=2 (Ours)** | | | | | | | | | | | | | | |
| Projector | **12.62** | 0.393 | 0.490 | 0.468 | 0.316 | 91.38 | 65.72 | 64.51 | 64.97 | 82.22 | 97.42 | 43.17 | 21.51 | 66.36 |
| **Encoder** | 15.52 | 2.175 | 0.603 | 0.616 | 5.364 | **71.52** | **47.24** | **25.10** | 24.32 | 63.67 | 67.81 | **21.56** | **13.55** | **41.85** |
| **Width=2** | | | | | | | | | | | | | | |
| Projector | 13.48 | 0.320 | 0.667 | 0.376 | 0.493 | 89.88 | 66.46 | 64.51 | 53.40 | 82.50 | 95.77 | 41.76 | 23.90 | 64.77 |
| Encoder | 16.46 | 2.341 | 0.607 | 0.646 | 5.899 | 73.05 | 50.61 | 27.25 | 25.60 | 64.84 | 67.87 | 22.35 | 15.10 | 43.33 |
| **Detection Error ↓** | | | | | | | | | | | | | | |
| **Depth=1** | | | | | | | | | | | | | | |
| Projector | 12.86 | 0.375 | 0.649 | 0.500 | 1.157 | 80.15 | 95.98 | 81.68 | 84.18 | 92.75 | **98.38** | **73.62** | 92.24 | 87.37 |
| Encoder | 16.34 | 1.673 | 0.667 | 0.589 | 7.936 | 62.72 | 95.04 | 84.65 | 84.95 | 92.22 | 99.43 | 89.75 | 83.66 | 86.55 |
| **Depth=2 (Ours)** | | | | | | | | | | | | | | |
| **Projector** | **12.62** | 0.393 | 0.490 | 0.468 | 0.316 | **60.85** | **48.23** | **42.35** | **67.69** | **56.51** | 99.04 | 76.32 | **69.84** | **65.10** |
| Encoder | 15.52 | 2.175 | 0.603 | 0.616 | 5.364 | 67.17 | 98.14 | 81.76 | 84.95 | 84.57 | 99.70 | 97.36 | 87.34 | 87.62 |
| **Width=2** | | | | | | | | | | | | | | |
| Projector | 13.48 | 0.320 | 0.667 | 0.376 | 0.493 | 65.43 | 60.83 | 51.96 | 67.77 | 57.70 | 99.52 | 79.29 | 75.33 | 69.73 |
| Encoder | 16.46 | 2.341 | 0.607 | 0.646 | 5.899 | 66.80 | 97.64 | 89.61 | 83.42 | 88.89 | 98.89 | 98.58 | 94.39 | 89.78 |

Table 20: **Comprehensive Results with SGD Optimizer.** VGG17 models are trained on **ImageNet-100** dataset (ID) using the SGD optimizer. Baseline models do not incorporate mechanisms like entropy regularization or the ETF projector to control NC. A lower NC value indicates stronger neural collapse. For OOD transfer we report $\mathcal{E}_{\text{GEN}}$ (%) whereas for OOD detection we report $\mathcal{E}_{\text{DET}}$ (%). OOD performance is averaged across eight OOD datasets. The same color highlights the rows to compare.

| Model | $\mathcal{E}_{\text{ID}}$↓ IN 100 | Neural Collapse | | | | OOD Datasets | | | | | | | | |
|---|---|---|---|---|---|---|---|---|---|---|---|---|---|---|
| | | $\mathcal{NC}1$ | $\mathcal{NC}2$ | $\mathcal{NC}3$ | $\mathcal{NC}4$ | IN-R 200 | CIFAR 100 | Flowers 102 | NINCO 64 | CUB 200 | Aircrafts 100 | Pets 37 | STL 10 | Avg. |
| **Transfer Error ↓** | | | | | | | | | | | | | | |
| **VGG17** | | | | | | | | | | | | | | |
| Encoder | **13.06** | 1.017 | 0.449 | 0.479 | 26.459 | 82.10 | **56.82** | 59.71 | 39.20 | 77.70 | 88.21 | 35.00 | 18.58 | 57.17 |
| **VGG17+Ours** | | | | | | | | | | | | | | |
| Projector | **13.18** | 0.087 | 0.468 | 0.267 | 0.264 | 84.48 | 64.69 | 72.35 | 47.28 | 85.88 | 94.66 | 42.85 | 25.93 | 64.77 |
| **Encoder** | 15.36 | 0.459 | 0.804 | 0.972 | 3.898 | **78.20** | 57.27 | **45.59** | **32.06** | **74.27** | **74.49** | **27.58** | **17.80** | **50.91** |
| **Detection Error ↓** | | | | | | | | | | | | | | |
| **VGG17** | | | | | | | | | | | | | | |
| Encoder | **13.06** | 1.017 | 0.449 | 0.479 | 26.459 | 81.63 | 93.34 | 87.25 | 86.31 | 94.89 | 97.93 | 84.76 | 91.40 | 89.69 |
| **VGG17+Ours** | | | | | | | | | | | | | | |
| **Projector** | **13.18** | 0.087 | 0.468 | 0.267 | 0.264 | **52.28** | **35.80** | **60.59** | **67.60** | **67.77** | **67.35** | **67.52** | **67.60** | **60.81** |
| Encoder | 15.36 | 0.459 | 0.804 | 0.972 | 3.898 | 93.58 | 95.11 | 68.43 | 84.01 | 87.42 | 87.16 | 87.00 | 86.48 | 86.15 |

Table 21: **Fixed ETF Classifier Vs. Plastic Classifier.** VGG17 models are trained on **ImageNet-100** dataset (ID) and evaluated on eight OOD datasets. All models incorporate entropy regularization and the ETF projector to control NC; **only the classifier (final layer) differs, being either trainable or a fixed ETF**. The same color highlights the rows to compare. All metrics except NC are reported in percentage. For OOD transfer we report $\mathcal{E}_{\text{GEN}}$ (%) whereas for OOD detection we report $\mathcal{E}_{\text{DET}}$ (%).

| Classifier (Last layer) | $\mathcal{E}_{\text{ID}}\downarrow$ IN 100 | $\mathcal{NC}1$ | $\mathcal{NC}2$ | $\mathcal{NC}3$ | $\mathcal{NC}4$ | IN-R 200 | CIFAR 100 | Flowers 102 | NINCO 64 | CUB 200 | Aircrafts 100 | Pets 37 | STL 10 | Avg. |
|---|---|---|---|---|---|---|---|---|---|---|---|---|---|---|
| **Transfer Error** ↓ | | | | | | | | | | | | | | |
| **Fixed ETF** | | | | | | | | | | | | | | |
| Projector | 13.56 | 0.088 | 0.702 | 0.374 | 0.379 | 98.18 | 84.28 | 92.25 | 96.94 | 96.86 | 97.60 | 72.23 | 36.59 | 84.37 |
| Encoder | 16.40 | 3.794 | 0.773 | 0.786 | 54.24 | 82.47 | 63.19 | 55.98 | 36.31 | 81.00 | 88.36 | 31.18 | 20.88 | 57.42 |
| **Plastic (Ours)** | | | | | | | | | | | | | | |
| Projector | **12.62** | 0.393 | 0.490 | 0.468 | 0.316 | 91.38 | 65.72 | 64.51 | 64.97 | 82.22 | 97.42 | 43.17 | 21.51 | 66.36 |
| **Encoder** | 15.52 | 2.175 | 0.603 | 0.616 | 5.364 | **71.52** | **47.24** | **25.10** | **24.32** | **63.67** | **67.81** | **21.56** | **13.55** | **41.85** |
| **Detection Error** ↓ | | | | | | | | | | | | | | |
| **Fixed ETF** | | | | | | | | | | | | | | |
| Projector | 13.56 | 0.088 | 0.702 | 0.374 | 0.379 | 73.80 | **26.45** | 73.04 | 68.20 | **55.80** | 98.98 | 96.05 | **63.56** | 69.49 |
| Encoder | 16.40 | 3.794 | 0.773 | 0.786 | 54.24 | 81.03 | 98.98 | 81.57 | 87.25 | 97.29 | 99.01 | 86.48 | 93.11 | 90.59 |
| **Plastic (Ours)** | | | | | | | | | | | | | | |
| **Projector** | **12.62** | 0.393 | 0.490 | 0.468 | 0.316 | **60.85** | 48.23 | **42.35** | 67.69 | 56.51 | 99.04 | **76.32** | 69.84 | **65.10** |
| Encoder | 15.52 | 2.175 | 0.603 | 0.616 | 5.364 | 67.17 | 98.14 | 81.76 | 84.95 | 84.57 | 99.70 | 97.36 | 87.34 | 87.62 |

Table 22: **Comprehensive Comparison with NECO.** We compare our method against a SOTA OOD detection method NECO (Ammar et al., 2024). Since NECO does not address OOD generalization, we do not compare OOD generalization performance. Here, various DNNs are trained on **ImageNet-100** dataset (ID) and evaluated on eight OOD datasets. NC metrics are computed using the penultimate-layer embeddings. A lower NC value indicates stronger neural collapse.

| Model | $\mathcal{E}_{\text{ID}}\downarrow$ IN 100 | $\mathcal{NC}1$ | $\mathcal{NC}2$ | $\mathcal{NC}3$ | $\mathcal{NC}4$ | IN-R 200 | CIFAR 100 | Flowers 102 | NINCO 64 | CUB 200 | Aircrafts 100 | Pets 37 | STL 10 | Avg. |
|---|---|---|---|---|---|---|---|---|---|---|---|---|---|---|
| **VGG17** | | | | | | | | | | | | | | |
| NECO | 12.18 | 0.766 | 0.705 | 0.486 | 37.491 | 83.20 | **26.41** | 87.94 | 78.74 | 83.45 | 83.53 | 96.76 | 82.49 | 77.82 |
| **Ours** | 12.62 | 0.393 | 0.490 | 0.468 | 0.316 | **60.85** | 48.23 | **42.35** | **67.69** | **56.51** | 99.04 | **76.32** | **69.84** | **65.10** |
| **ResNet18** | | | | | | | | | | | | | | |
| NECO | 15.38 | 1.11 | 0.658 | 0.590 | 31.446 | 93.88 | 66.81 | 94.12 | 90.05 | 90.05 | **90.05** | 90.05 | 90.05 | 88.13 |
| **Ours** | 16.14 | 0.341 | 0.456 | 0.306 | 0.540 | **67.92** | **61.21** | **71.18** | **71.09** | **23.20** | 99.28 | **81.41** | **82.29** | **69.70** |
| **ViT-T** | | | | | | | | | | | | | | |
| NECO | 31.78 | 2.467 | 0.657 | 0.601 | 1.015 | **79.90** | 74.76 | 82.84 | 84.44 | 85.12 | **98.50** | **92.67** | **87.10** | 85.67 |
| **Ours** | 32.04 | 2.748 | 0.609 | 0.798 | 1.144 | 81.12 | **60.81** | **77.55** | **82.40** | **79.05** | 99.10 | 95.15 | 90.06 | **83.16** |

Table 23: **Compute Overhead.** We compare our method with baseline DNNs in terms of FLOPs and training time. Training time (wall-clock) is reported in minutes.

| Model | FLOPs Comparison | | Time Comparison | |
|---|---|---|---|---|
| | FLOPs ↓ | % Increase | Time (Mins) ↓ | % Increase |
| VGG17 | 4,955,622,740,132,864 | – | 307.80 | – |
| **VGG17 + Ours** | 4,956,972,705,684,480 | **+0.0272%** | 307.98 | **+0.0585%** |
| ResNet18 | 461,500,110,825,472 | – | 136.81 | – |
| **ResNet18 + Ours** | 462,031,742,464,000 | **+0.1152%** | 137.04 | **+0.1681%** |
| ResNet34 | 931,123,885,195,264 | – | 140.89 | – |
| **ResNet34 + Ours** | 931,655,516,833,792 | **+0.0571%** | 141.26 | **+0.2626%** |
| ViT-T | 271,301,725,913,088 | – | 63.03 | – |
| **ViT-T + Ours** | 271,376,414,539,776 | **+0.0275%** | 63.18 | **+0.2380%** |
| ViT-S | 1,068,921,092,308,992 | – | 102.02 | – |
| **ViT-S + Ours** | 1,069,219,652,567,040 | **+0.0279%** | 102.24 | **+0.2156%** |

