# OpenReview forum: "Controlling Neural Collapse Enhances Out-of-Distribution Detection and Transfer Learning"
_ICML.cc/2025/Conference — ICML 2025 poster_

### Official Review · Reviewer_Cfr3 · 2025-02-21

**Overall Recommendation:** 3

**Summary:**

This paper explores the relationship between Neural Collapse, Out-of-Distribution detection, and OOD generalization. It provides that strong NC enhances OOD detection but damages generalization, while weaker NC has the opposite effect. In order to balance these objectives, the authors use a method that controls NC at different neural network layers.

**Claims And Evidence:**

The paper claims that NC has an inverse relationship with OOD detection and generalization and that controlling NC at different layers improves both objectives. These claims are supported by empirical evidence demonstrating:

- strong correlation between NC and OOD detection -- generalization performance
- improved results across multiple datasets when NC is manipulated using entropy regularization and a simplex ETF projector
- theoretical justification linking entropy regularization to preventing NC + demonstrating the benefits of an ETF projector for OOD detection

**Essential References Not Discussed:**

The paper appears to be covering most relevant prior work

**Experimental Designs Or Analyses:**

The experiments appears to be well-structured and executed, with comprehensive comparisons to baselines and ablation studies. The encoder with entropy regularization maintains feature diversity for better OOD generalization. The projector with a fixed ETF structure enhances OOD detection by inducing NC and the projector, _o_ in _f-o-g_, structure appears to be well evaluated. he authors also examine the hyperparameter and architectural sensitivity.

**Methods And Evaluation Criteria:**

The proposed method appears to be well-motivated and aligns with the problem at hand. Comparisons with baseline methods are performed on standard OOD benchmarks (e.g., NINCO, CIFAR-100, ImageNet-R). Metrics such as OOD detection error, OOD generalization error, and NC properties (NC1-NC4) are used. A range of architectures (VGG17, ResNet18, ViT) to ensure robustness of findings across varying architectures of different nature.

**Other Comments Or Suggestions:**

No comments

**Other Strengths And Weaknesses:**

- Providing a detailed analysis of computational efficiency and trade-offs could benefit the evaluation of the proposed framework
- Discussing potential limitations in scaling the method to deeper architectures wrt the NC could enhance clarity

**Questions For Authors:**

- What is the computational overhead of the proposed method compared to standard classifiers?
- Can this approach be extended to other tasks such as detection, i.e., is the proposed approach agnostic to the task at hand?

**Relation To Broader Scientific Literature:**

The paper builds on literature in NC, OOD detection, and transfer learning and cites key works and recent advances in ETF-based learning. It extends prior studies by linking NC explicitly to both detection and generalization tasks and proposing a practical mecanism to control NC at different network layers.

**Theoretical Claims:**

The theoretical component of the paper appears to be solid, with well-executed derivations explaining how entropy regularization mitigates NC and how a Simplex ETF projector enforces NC. The proofs seem to be accurate -- though further clarity in some steps of the entropy collapse argument could improve readability.

---

> ### Author Rebuttal · Authors · 2025-03-27
>
> Thank you for your thoughtful review and valuable feedback. We have carefully considered your concerns and tried to address them. Below, we have provided detailed responses to each review separately.
>
> # Weaknesses
>
> **W1. Is the proposed method computationally efficient?**
>
> Yes, the proposed method is computationally efficient. It introduces two additional components compared to standard DNN architecture and training protocol:
>
> 1. Entropy regularization applied at the encoder’s output.
> 2. A frozen ETF projector (two MLP layers) following the DNN backbone.
>
> - For *entropy regularization*, we employ an efficient batch-level nearest neighbor distance computation, which incurs negligible computational overhead during training.
> - Regarding the *ETF projector*, since it remains frozen and does not undergo gradient updates, it does not introduce any noticeable training costs beyond those of the baseline DNN.
>
> To quantify this, when training DNNs (VGG17, ResNet18/34, ViT-T/S) on ImageNet-100 (ID dataset) for 100 epochs using four NVIDIA RTX A5000 GPUs, both our method and the baseline require the same training time. Moreover, in terms of FLOPs (floating-point operations per second), both our method and the baseline require the same amount of computation. This indicates that our method does not require higher computational costs than standard DNNs. Below, we report the FLOPs and training time:
>
> | Model | Compute (FLOPs $\times 10^{15}$) | Time (Hours) |
> |----------|----------|----------|
> | VGG17   | 4.96   | 5.13   |
> | VGG17+Ours   | 4.96   | 5.13   |
> | ResNet18   | 0.46   | 2.28   |
> | ResNet18+Ours   | 0.46   | 2.28   |
> | ResNet34   | 0.93   | 2.35   |
> | ResNet34+Ours   | 0.93   | 2.35   |
> | ViT-T   | 0.27   | 1.05   |
> | ViT-T+Ours   | 0.27   | 1.05   |
> | ViT-S   | 1.07   | 1.70   |
> | ViT-S+Ours   | 1.07   | 1.70   |
>
> We will include this comparison in the final paper and appreciate the reviewer’s suggestion to provide this computational analysis.
>
>
> **W2. Does the proposed method scale to deeper architectures?**
>
> - Our method is inherently compatible with deeper architectures since the ETF projector (two MLP layers) can be seamlessly integrated into encoders of any depth. Additionally, while deeper DNNs typically exhibit stronger Neural Collapse (NC) in their top layers, our entropy regularizer effectively mitigates NC in encoders of any depth.
> - As demonstrated in Table 9 (Appendix), our method performs effectively with both ResNet-18 (13M) and ResNet-34 (23M), highlighting its scalability. Also, as shown in Table 10 (Appendix), our method improves OOD detection and generalization performance for both ViT-Tiny (6M) and ViT-Small (23M), demonstrating its scalability.
>
>
> # Questions
>
> **Q1. What is the computational overhead of the proposed method compared to standard classifiers?**
>
> As detailed in W1, our method does not increase computational overhead compared to standard DNNs.
>
>
> **Q2. Can this approach be extended to other tasks such as detection, i.e., is the proposed approach agnostic to the task at hand?**
>
> Yes, the proposed method is agnostic to the task at hand. Although we primarily focus on classification tasks, our method is applicable to other tasks, e.g., objection detection or other regression tasks. Our method employs an ETF projector after the DNN backbone (encoder). This ETF projector is designed to increase Neural Collapse in the last layer, thereby improving feature representations for downstream tasks, including classification and regression tasks. Our entropy regularization is also task-agnostic and compatible with both classification and regression objectives. We will discuss this in the main text. We thank the reviewer for the insightful questions.
>
> We appreciate the reviewer's insightful feedback and suggestions. Please let us know if further clarifications are needed.

---

> > ### Comment · Reviewer_Cfr3 · 2025-04-08
> >
> > I thank the reviewers for addressing my comments and appreciate the additional study showing the computational overhead in terms of wall time. I suggest adding this to the paper with higher point precision, as the overhead is not observable with the current precision. Apart from that, I have no other concerns. I would keep my original score.

---

> > > ### Author Response · Authors · 2025-04-08
> > >
> > > We thank the reviewer for the constructive feedback.
> > > As suggested, we report the computational overhead of our method using higher-precision measurements, presented both in terms of FLOPs and training time. The percentage increase introduced by our method is *minimal* across all architectures, as shown below:
> > >
> > > ---
> > >
> > > # FLOPs Comparison
> > >
> > > | Model         | FLOPs                | Delta (% Increase) |
> > > |---------------|----------------------|---------------------|
> > > | VGG17         | 4,955,622,740,132,864 |                    |
> > > | VGG17+Ours    | 4,956,972,705,684,480 | **+0.0272%**        |
> > > | ResNet18      | 461,500,110,825,472  |                     |
> > > | ResNet18+Ours | 462,031,742,464,000  | **+0.1152%**        |
> > > | ResNet34      | 931,123,885,195,264  |                     |
> > > | ResNet34+Ours | 931,655,516,833,792  | **+0.0571%**        |
> > > | ViT-T         | 271,301,725,913,088   |                    |
> > > | ViT-T+Ours    | 271,376,414,539,776   | **+0.0275%**        |
> > > | ViT-S         | 1,068,921,092,308,992 |                    |
> > > | ViT-S+Ours    | 1,069,219,652,567,040 | **+0.0279%**        |
> > >
> > > ---
> > >
> > > # Training Time Comparison
> > >
> > > | Model         | Time (minutes) | Delta (% Increase) |
> > > |---------------|----------------|---------------------|
> > > | VGG17         | 307.80         |                     |
> > > | VGG17+Ours    | 307.98         | **+0.0585%**        |
> > > | ResNet18      | 136.81         |                     |
> > > | ResNet18+Ours | 137.04         | **+0.1681%**        |
> > > | ResNet34      | 140.89         |                     |
> > > | ResNet34+Ours | 141.26         | **+0.2626%**        |
> > > | ViT-T         | 63.03          |                     |
> > > | ViT-T+Ours    | 63.18          | **+0.2380%**        |
> > > | ViT-S         | 102.02         |                     |
> > > | ViT-S+Ours    | 102.24         | **+0.2156%**        |
> > >
> > > ---
> > >
> > > As shown, the overhead introduced by our method remains below **0.3%** in all cases, which we believe is trivial and well-justified given the observed performance gains. We have included this computational analysis in our revised paper. We thank the reviewer for the valuable suggestions, which have helped strengthen our paper.
> > >
> > > We hope this addresses the remaining concerns. We are grateful for your constructive feedback and would appreciate it if you could reconsider the final score of our submission.

---

### Official Review · Reviewer_WkZc · 2025-02-23

**Overall Recommendation:** 3

**Summary:**

This paper empirically shows a trade-off between Out-of-distribution (OOD) detection and OOD generalization on multi-class classification task in the deep neural network (DNN) with in-distribution (ID) training data: a network can either impose a stronger neural collapse (NC) and improves OOD detection or weakens NC to enhances OOD generalization.

To enhance both aspects, this paper proposes a pipeline to train the backbone for OOD generalization and add a projector and linear probing after it for OOD detection, hence the backbone which encodes the embeddings only needs to be trained once. (See Figure 2 in the paper)

To modify the architecture to the transfer learning setting, this paper uses a novel *entropy regularization*, which mitigates NC,
and replaces batch-normalization (BN) by group normalization (GN) plus weight standardization (WS). These changes further improves OOD detection and generalization performance.

**Claims And Evidence:**

The claims are supported clearly and evidently by various experiments and ablation tests.

**Essential References Not Discussed:**

I think most of the essential related works are cited. A few more minor but related work which might be worth to cite can be:

[4]: also about OOD and NC. Would be interesting to see which method could yield better performance.

[5]: in contrary to line 421-422, "How BN or GN impacts neural collapse is unexplored in previous work.", BN has been explored in [5].

----------

[4] Ammar, Mouïn Ben, et al. "Neco: Neural collapse based out-of-distribution detection." arXiv preprint arXiv:2310.06823 (2023).

[5] Pan, Leyan, and Xinyuan Cao. "Towards understanding neural collapse: The effects of batch normalization and weight decay." arXiv preprint arXiv:2309.04644 (2023).

**Experimental Designs Or Analyses:**

While I generally acknowledge the validity of the experiments, I would like to point out that:

1. The datasets used are all vision tasks. It would be more complete if the paper could draw similar conclusions on OOD detections/generalizations for other tasks with other input forms like audio or text.

2. This paper only used AdamW as the optimisers, while previous work, to name a few, [1,2,3], mainly used SGD or Adam, to showcase neural collapse. I personally also experiment on neural collapse (with the same architectures but datasets smaller than those in this paper), and find out that AdamW does not show neural collapse. For example, in Table 4, the NC metrics NC2, NC3 are about 0.5, which are significantly away from zero, since the metric is normalised between 0 and 2. If SGD or Adam is used as optimisers, the metrics could drop to the magnitude by 1 or 2. I still believe the novel findings in this paper, like entropy regularization and encoder-projector-classifier pipeline, are significant works on OOD context, but since the title includes the term neural collapse, we should be careful on the experiments and NC metrics.


---
[1] Papyan, V., Han, X., and Donoho, D. L. Prevalence of neural collapse during the terminal phase of deep learning training. Proceedings of the National Academy of Sciences, 117(40):24652–24663, 2020.

[2] Zhu, Z., Ding, T., Zhou, J., Li, X., You, C., Sulam, J., and Qu, Q. A geometric analysis of neural collapse with unconstrained features. Advances in Neural Information Processing Systems, 34:29820–29834, 2021.

[3] Masarczyk, W., Ostaszewski, M., Imani, E., Pascanu, R., Miło´s, P., and Trzcinski, T. The tunnel effect: Building data representations in deep neural networks. Advances in Neural Information Processing Systems, 36, 2023.

**Methods And Evaluation Criteria:**

The paper uses 8 commonly used OOD datasets, 3 DNN architectures trained on AdamW with cosine learning rate scheduler and linear warmup. (See Section 5)

I think it is rather standard for vision OOD detection/generalization task.

**Other Comments Or Suggestions:**

Just a minor stuff, I would suggest to flip the axes of Figure 4: the dependent variable is NC1, which is usually shown on the y-axis.

**Other Strengths And Weaknesses:**

The contributions of this paper seem original and it is delivered in a clear way. The experiments' description is very detailed.

**Questions For Authors:**

Regarding my concerns on the experimental designs, I would like to ask:

1. Could the paper include non-vision datasets in the revised edition to show that the contribution is universal among different data type?

2. Could the experiment be repeated with different optimisers like SGD or Adam to see if the the effect is intensified, neutralised or unchanged on OOD performance?

---

regarding Figure 1 and 4, I would also like to know:

3. why is there a gap between NC1 in the middle of the plots where no instance can achieve NC1 around 0.5? Is there an explanation for this or could it be a bug in the code?

---

regarding my concern in theoretical claims,

4. could the paper include a formal proof of Prop. 4.1?

**Relation To Broader Scientific Literature:**

This paper clearly relates to the neural collapse phenomenon introduced by [1]. I am not an expert in OOD and transfer settings so I cannot draw more conclusions on this.

---
[1] Papyan, V., Han, X., and Donoho, D. L. Prevalence of neural collapse during the terminal phase of deep learning training. Proceedings of the National Academy of Sciences, 117(40):24652–24663, 2020.

**Theoretical Claims:**

There is only one theoretical claim, namely Proposition 4.1, with proof sketch in the main text which promises a detailed proof in Appendix E. While in Appendix E, the proof is still rather sketchy. I still believe the proposition is true, but I hope the proof can be written in a more formal way in the revised edition.

---

> ### Author Rebuttal · Authors · 2025-03-27
>
> Thank you for your thoughtful reviews and constructive feedback. We have carefully considered your concerns and tried to address them. Below, we have provided detailed responses to each review separately.
>
> # Clarifications
>
> **Non-vision datasets or tasks**
>
> Most prior work on OOD detection and/or OOD generalization has focused on vision datasets, and we align with this precedent. However, we recognize the importance of evaluating non-vision tasks for broader applicability. Due to time and computational constraints, we were unable to extend our work to non-vision domains within the rebuttal period. We consider this an important direction for future work and will discuss potential extensions to other modalities, such as audio and text.
>
> **Code and reproducibility**
>
> At the time of submission, our code was not fully polished, which is why we did not include it. However, we plan to release a cleaned and well-documented version upon acceptance.
>
> **Performance with optimizers other than AdamW**
>
> We have addressed this concern in Q2.
>
> # Additional Suggestions
>
>  - We have revised the statement ``How BN or GN impacts Neural Collapse is unexplored in prior work." since BN has been studied in previous work [1]. We thank the reviewer for pointing this out.
>
>  - **Essential References Not Discussed:** We have cited the relevant papers [1, 2] suggested by the reviewer. We have included NECO [2] in our related work and discussed its relevance. We thank the reviewer for the valuable feedback.
>
>   - **References:**
>  1. Pan, Leyan, and Xinyuan Cao. "Towards understanding neural collapse: The effects of batch normalization and weight decay." arXiv preprint arXiv:2309.04644 (2023).
>  2.  Ammar, Mouïn Ben, et al. "Neco: Neural collapse based out-of-distribution detection." ICLR, 2024.
>
>  - **Comparison with NECO:**  NECO [2] is a SOTA OOD detection method based on Neural Collapse. To compare against NECO, we trained various DNNs on ImageNet-100 (ID) and evaluated them on eight OOD datasets. Our method outperforms NECO in average OOD detection error by an absolute 12.72\% for VGG17, 18.43\% for ResNet18, and 2.51\% for ViT-T. More details are given in responses to reviewer BRBv. We have included these results in our paper. We thank the reviewer for the valuable suggestions.
>
>  - We have improved the clarity of Figure 4 by flipping the axes, as per the reviewer’s suggestion.
>
> # Questions
>
> **Q1. Could authors include non-vision experiments?**
>
> We have addressed this concern in the clarifications section above. We welcome any further thoughts on this.
>
> **Q2. Could authors perform experiments with SGD or Adam Optimizer?**
>
> Running all experiments with SGD or Adam would be computationally expensive due to our extensive evaluation across five DNN architectures and eight OOD datasets. However, we conducted ablation studies by training VGG17 models with an SGD optimizer and evaluating them on eight OOD datasets.
> Our SGD results confirm that our method consistently improves OOD detection and generalization performance.
> - In particular, our method outperforms the baseline by an absolute 6.26\%
> in OOD generalization and by an absolute 28.88\% in OOD detection.
> - Also, we observe that our encoder reduces Neural Collapse (NC) and enhances OOD generalization by an absolute 13.86\% compared to the projector. Whereas the projector intensifies NC and improves OOD detection by an absolute 25.34\% compared to the encoder.
> - While SGD intensifies NC more than AdamW, AdamW achieves better overall performance (Table 16 in Appendix).
>
> **SGD Optimizer Results:**
>
> | Model | $\mathcal{E}_{\text{ID}}$ | NC1 | NC2 | NC3 | NC4 | $\mathcal{E}_{\text{GEN}}$ | $\mathcal{E}_{\text{DET}}$ |
> |----------|----------|----------|----------|----------|----------|----------|----------|
> | VGG17 | $13.06$ | $1.02$ | $0.45$ | $0.48$ | $26.46$ | $57.17$ | $89.69$ |
> | +Ours   | $13.18$ | $0.09$ | $0.47$ | $0.27$ | $0.26$  | $\mathbf{50.91}$ | $\mathbf{60.81}$ |
>
> We have included these results in our paper. We will try to conduct more experiments with SGD/ Adam and include the results in the final paper. We would appreciate the reviewer’s feedback on whether this analysis sufficiently addresses the concern.
>
> **Q3. Why is there a gap in correlation figures?**
>
> We observe this trend for VGG17 but not for ResNet18 (Fig. 7 in Appendix). We have double-checked our data and code, they seem fine. The potential reason could be the limited data points, typically more data points would show better trends. It might be better if we combine Fig. 1 and Fig. 7.
>
> **Q4. Could the paper include a formal proof of Proposition 4.1?**
>
> Yes, we have included a detailed formal proof of Proposition 4.1 in Appendix E. We thank the reviewer for the valuable feedback and insightful questions.
>
> We appreciate the reviewer's insightful feedback and suggestions. Please let us know if further clarifications are needed.

---

> > ### Comment · Reviewer_WkZc · 2025-04-05
> >
> > Thank you for your detailed answer. I have no further questions.
> >
> > I would like to maintain my original scoring.

---

> > > ### Author Response · Authors · 2025-04-05
> > >
> > > We thank the reviewer for their valuable feedback, which has helped improve this paper. If our responses have adequately addressed your concerns, we humbly seek your support for the paper.

---

### Official Review · Reviewer_BRBv · 2025-03-05

**Overall Recommendation:** 3

**Summary:**

This paper studies the role of neural collapse on the out-of-distribution (OOD) detection and generalization tasks. The authors also propose an entropy regularization technique to control the degree of collapse in intermediate layers and show that stronger collapse in the final layers can aid in OOD detection but can hurt OOD generalization.

## update after rebuttal

The authors have addressed my concerns and have also assured that the code will be open-sourced. Overall I appreciate the extra experiments that they did during the rebuttal to further justify their approach (when compared to previous works). I also increased my score from 2 -> 3 after the rebuttal.

**Claims And Evidence:**

Experiments and analysis with various model architectures and datasets justify the empirical claims (some concerns are mentioned in the following sections). However, the authors do not justify their claims about a "theoretical framework explaining how a fixed Simplex ETF projector enforces NC for OOD detection". This claim must be revised to state that such an analysis is only experimental.

**Essential References Not Discussed:**

1. A relevant previous effort on NC and OOD-Generalization [1] is not discussed in the paper. Can the authors provide some comparisons with this paper ?

2. I have noticed that in Section 6.5 (in Appendix), the authors state that "How BN or GN impacts neural collapse is unexplored in previous work". This is not valid as previous works such as [2] have indeed studied this aspect. It would be great if the authors can rectify this statement.

[1] Ammar, Mouin Ben, et al. "NECO: NEURAL COLLAPSE BASED OUT-OF-DISTRIBUTION DETECTION." The Twelfth International Conference on Learning Representations. 2024.

[2] Ergen, T., et al. "Demystifying Batch Normalization in ReLU Networks: Equivalent Convex Optimization Models and Implicit Regularization." International Conference on Learning Representations. 2022.

**Experimental Designs Or Analyses:**

The design of the experiments is valid and sound. However, I have some concerns with the reported numbers and would like the authors to clarify the following:

1. Why are the baseline VGG17 results for $\mathcal{E}_{ID}$ in Table 2 (first row) different from the non entropy regularization case in Table 3 (first row)? The captions do not indicate any major difference.

2. Next, the second row in Table 2 (i.e, the proposed method with VGG17 backbone + fixed ETFs and entropy regularization) has the same ID and OOD performance as a VGG17 model with entropy regularization in the second row of Table 3. Does the latter also employ the fixed ETFs as projectors? If yes, what is the difference between these two models? How can the ID and OOD performance be the same even though the NC values are different?

**Methods And Evaluation Criteria:**

The NC and evaluation metrics, datasets, and models are relevant to the analysis.

**Other Comments Or Suggestions:**

N/A

**Other Strengths And Weaknesses:**

The paper is generally well-written and easy to follow. However, a lack of comparison with previous work is a bit concerning. In particular, for a given OOD detection/generalization task, I am not convinced that a comparison cannot be done with any of the previous work in the literature. I would like to know the authors' justification on this aspect.

**Questions For Authors:**

1. The entropy regularization applied to a standard VGG17 without any fixed ETF layer weights is shown in Table 12. Is the regularization applied to the last layer in this experiment? If yes, then it is not a fair comparison with VGG17 + 2 layer ETF architecture since technically the regularization should be applied to the last but second/third layer (depending on the usage of classifier head). Can the authors clarify this?

**Relation To Broader Scientific Literature:**

This paper analyzes the role of neural collapse (NC) on OOD generalization and detection tasks in NNs. In the NC literature there has been extensive work on analyzing the optimality of collapse in intermediate and final layers and the implications on transfer learning and in-domain generalization. However, the study on OOD detection+generalization and NC is still in its nascent stage. This paper is well positioned to present an initial unifying perspective (at least empirically) and encourage future efforts in this direction.

**Theoretical Claims:**

The authors present the proof of their Proposition 4.1 in Appendix E. (on the entropy regularization tending to -inf with stronger collapse in the features).

---

> ### Author Rebuttal · Authors · 2025-03-27
>
> Thank you for your thoughtful reviews and valuable feedback. We have carefully considered your concerns and tried to address them. Below, we have provided detailed responses to each review separately.
>
> # Clarifications on Tables 2 and 3
>
>  - In Table 3, we isolate the effect of entropy regularization while keeping all other components identical. Specifically, we compare a baseline VGG17 with our method, where both models include the ETF projector, and only entropy regularization is either applied or omitted. In contrast, in Table 2, the baseline VGG17 does not include the ETF projector, which explains the differences in ID/ OOD performance between the first rows of the two tables.
>
>  - The second rows of Tables 2 and 3 correspond to the same model (VGG17 + ETF projector + entropy regularizer). The NC values differ because, in Table 2, we measure NC after the projector, whereas in Table 3, we report NC at the encoder. Since entropy regularization is applied to the encoder, Table 3 explicitly compares its effect on encoder representations.
>  Table 1 reports NC values for both the encoder and the projector (first two rows), which remain consistent with Tables 2 and 3.
>
>  We will add more details to the table captions to enhance clarity. Please let us know if further clarification is needed.
>
> # Comparison with NECO
>
> We have included NECO [1] in our related work and discussed its relevance. NECO is a *post-hoc OOD detection method* based on Neural Collapse, requiring a pre-trained DNN. It does not focus on OOD generalization and representation learning. In contrast, our method is designed to learn good representations during training to improve both OOD detection and generalization.
>
> Reference: [1] Ammar et al., NECO: Neural collapse based out-of-distribution detection, ICLR, 2024
>
> **Our Method Vs. NECO:** To compare against NECO, we trained various DNNs on ImageNet-100 (ID) and evaluated them on eight OOD datasets. Our method outperforms NECO in average OOD detection error by an absolute 12.72\% for VGG17, 18.43\% for ResNet18, and 2.51\% for ViT-T.
>
> **Below, we report the avg. OOD detection error (\%):**
> | Method | VGG17 | ResNet18 | ViT-T |
> |----------|----------|----------|----------|
> | NECO | $77.82$ | $88.13$ | $85.67$ |
> | Ours | $\mathbf{65.10}$ | $\mathbf{69.70}$ | $\mathbf{83.16}$ |
>
> We appreciate the reviewer’s suggestion and have incorporated the results in our paper.
>
> # Additional Suggestions
>
> - We have revised our statement about the theoretical framework explaining how a fixed simplex ETF projector enforces NC for OOD detection. We have stated that this analysis is empirical. We thank the reviewer for the valuable feedback.
>
> - We have revised this statement ``How BN or GN impacts neural collapse is unexplored in previous work." since BN has been studied in prior work [2]. We thank the reviewer for pointing this out.
>
> - Reference: [2] Ergen, T., et al. ``Demystifying Batch Normalization in ReLU Networks: Equivalent Convex Optimization Models and Implicit Regularization." ICLR, 2022.
>
> # Additional Weaknesses
>
> **Why is comparison with prior works difficult?**
>
> - Prior works typically focus on either OOD detection or OOD generalization, whereas we aim to improve both objectives. Some recent works address both objectives, but they differ significantly in problem formulation and methodology. As detailed in Sec. 3 and 5, we cannot directly compare against such methods because:
>
> 1. They rely on additional OOD training data, whereas our method works without such data. Adapting these methods for comparison would require substantial modifications, including redefined training objectives and architectures, which are beyond the scope of our work. Therefore, a direct *apple-to-apple* comparison is difficult.
>
> 2. For OOD generalization, prior work *solely* focuses on *co-variate shifted OOD* data that has similar label space as ID data (e.g., a car in *sunny weather* as ID vs. a car in *rainy weather* as OOD). In contrast, we focus on *semantic shifted OOD* data that does not overlap with ID labels. Thus, unlike our method, previous methods are not designed to handle semantic OOD data.
>
> We would appreciate the reviewer’s feedback on whether this justification sufficiently addresses the concern.
>
> # Questions
>
> - **Clarifications on Table 12:** Table 12 examines the impact of entropy regularization on the OOD generalization of a regular VGG17 without any projector. The regularization is applied to the last encoder layer (penultimate layer) before the final classifier layer. All models are compared identically. We are not sure if we understand the last part of the question relating to the VGG+ETF projector because we are not comparing it in Table 12.
>
> - Additionally, in Table 11, we compare how entropy regularization impacts OOD generalization when VGG17 models include ETF projectors.
>
> We appreciate the reviewer's insightful feedback and suggestions. Please let us know if further clarifications are needed.

---

> > ### Comment · Reviewer_BRBv · 2025-04-04
> >
> > Thanks for addressing most of my concerns.
> >
> > 1. I appreciate the authors' providing some baseline comparisons with NECO, as this helps readers connect your paper with the broader literature. I also see that the authors have not provided any code for reproducibility. Can they comment on this? Will the code be open-sourced?
> >
> > 2. Based on the author's response, I referred to [1] (as mentioned in section 3) and saw that even though they open-sourced the code, it is still a work in progress. On the other hand, [2] also open-sourced their code and seems to be up to date. The important thing to note here is that Table 1 in [2] considers sematic OOD data for OOD generalization experiments. Thus, in the "Differences from prior work" paragraph, I request the authors to carefully update the sentence formation so as to not give a wrong impression that previous works have only considered "covariate-shifted OOD" data.
> >
> > 3. Most importantly, thanks for acknowledging that the captions must be updated. For instance, it would be better if one could read the caption in Figure 11 and clearly understand that the VGG models include projectors.
> >
> > I do not have any further concerns and have increased the score. Good luck.
> >
> >
> > [1] Zhang, Q., Feng, Q., Zhou, J. T., Bian, Y., Hu, Q., and Zhang, C. The best of both worlds: On the dilemma of out-of-distribution detection. In Advances in Neural Information Processing Systems, 2024.
> >
> > [2] Wang, H. and Li, Y. Bridging ood detection and generalization: A graph-theoretic view. Advances in Neural
> > Information Processing Systems, 2024.

---

> > > ### Author Response · Authors · 2025-04-05
> > >
> > > We thank the reviewer for the valuable suggestions and for raising the score. Your feedback has helped improve our paper.
> > >
> > > - Regarding the code, we will open-source it upon acceptance to ensure full transparency and facilitate future research. The released repository will include scripts to reproduce all key experiments and results reported in the paper.
> > >
> > > - Regarding prior work, we acknowledge the reviewer’s observation about the types of OOD data used. We will revise the "Differences from prior work" paragraph to incorporate your suggestions.
> > >
> > > Thank you again for your insightful reviews and helpful feedback.

---

### Official Review · Reviewer_NGXx · 2025-03-10

**Overall Recommendation:** 4

**Summary:**

The paper proposes a novel claim that: stronger NC(neural collapse) improves OOD detection but degrades generalization, while weaker NC enhances generalization at the cost of detection. The explanation of above claim is sufficient and the experiment proves the statement.

**Claims And Evidence:**

Yes

**Essential References Not Discussed:**

The related work is sufficiently listed and discussed.

**Experimental Designs Or Analyses:**

The experiments are conducted on ImageNet-100 with different architectures. The balance of generalization and detection performance is clear and verify their statement about the generalization-detection trade-off.

**Methods And Evaluation Criteria:**

Yes

**Other Comments Or Suggestions:**

None

**Other Strengths And Weaknesses:**

The paper is well organized and expressed, and easy to understand.

**Questions For Authors:**

None

**Relation To Broader Scientific Literature:**

None

**Theoretical Claims:**

I do not check proofs for theoretical claims.

---

> ### Author Rebuttal · Authors · 2025-03-27
>
> Thank you for your thoughtful reviews and valuable feedback. Please let us know if you have any questions or suggestions. We would be happy to address them.

---

### Decision · Program_Chairs · 2025-05-01

**Decision:**

Accept (poster)

**Comment:**

This paper studies the relationship between Neural Collapse (NC) and out-of-distribution (OOD) detection and generalization. It demonstrates that strong NC improves OOD detection but harms generalization, while weaker NC has the opposite effect. To balance these objectives, the authors then propose a method to control NC across different neural network layers.

The idea of fixing classifiers to induce NC and the impact of NC on OOD generalization have been explored before, but this paper makes a valuable contribution by leveraging NC to improve both OOD detection and generalization, and by demonstrating the effectiveness of controlling NC in achieving these goals.